# Evaluating downscaled products with expected hydroclimatic co-variances

**Authors:** Seung H. Baek[1*], Paul A. Ullrich[1], Bo Dong[1], Jiwoo Lee[1]

**Affiliations:**

[1]Atmospheric, Earth, & Energy Division, Lawrence Livermore National Laboratory, Livermore, California USA

*Corresponding author. Email: baek1@llnl.gov

**Abstract:** There has been widespread adoption of downscaled products amongst practitioners and stakeholders to ascertain risk from climate hazards at the local scale (*e.g.,* ~5 km resolution). Such products must nevertheless be consistent with physical laws to be credible and of value to users. Here we evaluate statistically and dynamically downscaled products by examining local co-evolution of downscaled temperature and precipitation during convective and frontal precipitation events (two mechanisms testable with just temperature and precipitation). We find that two widely-used statistical downscaling techniques (LOCalized Analogs version 2 (LOCA2) and Seasonal Trends and Analysis of Residuals Empirical-Statistical Downscaling Model (STAR-ESDM)) generally preserve expected covariances during convective precipitation events over the historical and future projected intervals as compared to European Centre for medium-Range Weather Forecasts Reanalysis v5 (ERA5) and two observation-based data products (Livneh and nClimGrid-Daily). However, both techniques dampen future intensification of frontal precipitation that is otherwise robustly captured in global climate models (*i.e.,* prior to downscaling) and with process-based dynamical downscaling across five different regional climate models. In the case of LOCA2, this leads to appreciable underestimation of future frontal precipitation event intensity. This study is one of the first to quantify a likely ramification of the stationarity assumption underlying statistical downscaling methods and identify a phenomenon where projections of future change diverge depending on data production method employed. Finally, our work proposes expected covariances during convective and frontal precipitation as useful evaluation diagnostics that can be applied universally to a wide range of statistically downscaled products.

## 1. Introduction:

Extreme weather events are among the costliest disasters to the United States. Over the past four decades (1980-2023), there have been more than 370 billion-dollar disasters that cumulatively cost over 2.6 trillion dollars (NOAA, 2023). To ascertain risk from climate hazards, a broad community of practitioners, stakeholders and policymakers rely on historical reconstructions and future projections of local to regional climate that are "downscaled" from coarse global climate model outputs (Fiedler et al. 2021, Pitman et al. 2022). This is because global climate model (GCM) data alone are too coarse in resolution: GCM outputs from the Coupled Model Intercomparison Project 6 (CMIP6), for instance, have grid spacing of ~100 to 300 kilometers in the midlatitudes and cannot adequately represent finer scale features like topography and extreme storms (Eyring et al. 2016).

Numerous climate data products have emerged over the last several years that represent the contiguous United States (herein CONUS) at local scales, including dynamically and statistically downscaled products. Dynamically downscaled products (*e.g.,* Jones et al. 2022; Liu et al. 2017; Dai et al. 2020; Rasmussen et al. 2023; Chen et al. 2023) use regional climate models that simulate local meteorology, providing a comprehensive set of climate variables that are inherently self-consistent. While the general expectation is that dynamical downscaling implicitly preserves physical relationships among variables because they are generated by a modeling system based on physical laws, it is known that some biases can arise from insufficient representation of relevant physical processes (such as eddies; Xu et al. 2019), inherent error from lateral boundary input (*e.g.,* from GCMs; Rahimi et al. 2024), and/or sensitivity to regional climate model configurations (the limitations of regional climate models were comprehensively reviewed by Giorgi (2018) and Lloyd et al. (2021)) that must be considered. Statistically downscaled products (*e.g.,* Abatzoglou and Brown, 2012; Thrasher et al. 2012, 2022; Pierce et

al. 2014, 2023) are derived based on relationships between coarse climate model outputs and observed local meteorology (*e.g.,* Livneh et al. 2015; Durre et al., 2022). Since they are generated through simple functional relationships, statistically downscaled products can be generated more rapidly than dynamically downscaled products (albeit for fewer variables, as dense observational networks are only available for select quantities).

Given their computational convenience, there has been widespread adoption of statistically downscaled products. Statistically downscaled products must nevertheless be credible to be of value to users; the data must be consistent with physical laws to be trusted for future projections (Cash et al., 2002). Importantly, common statistical downscaling methods downscale variables independently of one another and thus do not explicitly account for covariances across variables at the local scale (notwithstanding existing covariances generated by climate models prior to downscaling). This may be problematic, as the loss of process-relevant covariances, if any, would undermine downstream assessments of multi-variate hazards (*e.g.,* droughts, flooding, and wildfires). Drought and wildfire metrics, for instance, may require self-consistent inputs of temperature and precipitation. Additionally, statistical downscaling assumes that observed functional relationships will be preserved in the future (*i.e.,* the stationarity assumption) despite climate change (Milly et al. 2008); however, there is no guarantee that historically derived statistical relationships will remain valid in the future. A precise understanding of the extent to which such assumption may undermine projections nevertheless remains elusive.

Here we assess the extent to which two locally relevant covariances between temperature and precipitation are preserved (or lost), as compared to outputs from global climate models and their dynamically downscaled counterparts, in two widely-used (*e.g.,* Martin, 2023; Ullrich, 2023; Jia et al. 2024; Najibi et al. 2024; Wang et al. 2024) statistical downscaling techniques

selected to accompany the Fifth National Climate Assessment (NCA5; the preeminent guidance

on national climate risks; USGCRP, 2023): LOCalized Analogs version 2 (LOCA2; Pierce et al.

2014) and Seasonal Trends and Analysis of Residuals Empirical-Statistical Downscaling Model

(STAR-ESDM; Hayhoe et al. 2024). A central goal of our paper is to understand the

representation of physical mechanisms in statistical downscaling products with only daily

surface temperature and precipitation outputs (often the only two variables available with

statistical downscaling). For this reason, we examine expected covariances between temperature

and precipitation during convective and frontal precipitation events, including for the projection

interval where the stationarity assumption may not hold. Although the credibility of both LOCA2

and STAR-ESDM has been evaluated for single variables (*e.g.,* Pierce et al., 2014; Hayhoe et al.,

2024), we propose for the first time diagnostics for evaluating *covariances* that can be applied

universally to a wide range of statistically downscaled products. Collectively, our work attempts

to address the following questions:

1) To what extent is physical consistency across variables preserved, as compared to

observations, when variables are (i) statistically downscaled independently and (ii)

dynamically downscaled concurrently?

2) How much does the stationarity assumption inherent in statistical downscaling

undermine credibility of projections, particularly for potentially non-stationary

hydrologic processes?

**2. Data and Methods:**

We employ outputs from eight Coupled Model Intercomparison Project Phase 6 (CMIP6)

models and their statistically downscaled counterparts (see Table 1). The statistically downscaled

products come from LOCalized Analogs version 2 (LOCA2; Pierce et al. 2014) and Seasonal Trends and Analysis of Residuals Empirical-Statistical Downscaling Model (STAR-ESDM; Hayhoe et al. 2024). The following description of LOCA2 and STAR-ESDM is from Ullrich (2023), with minor modifications. LOCA2 is a statistical downscaling technique based on signal decomposition employing analogs (*i.e.,* days in the historical record that exhibit regional meteorology most like the regional patterns of a given day in the CMIP6 model). The LOCA2 algorithm first bias-corrects historical CMIP6 outputs to observations using quantile mapping. It then adjusts the amount of variability seen in different frequency bands to match observations using a digital filter (Pierce et al., 2014). To downscale data at a given grid cell, the 30 days in the historical record best exhibiting regional meteorology as compared to the CMIP6 model day is identified. The single day best matching the model day is used as the analog for the local region around the grid point (Pierce et al., 2023). The LOCA2 North American product uses an updated version of Livneh et al. 2015 with 6-km grid spacing as the training dataset (Pierce et al. 2021). Outputs from LOCA2 are also available at 6-km grid resolution.

STAR-ESDM is a statistical downscaling technique based on signal decomposition (Hayhoe et al., 2023). The STAR-ESDM algorithm first disaggregates observations and GCM outputs into four separate components: the long-term trend, climatological annual cycle, annually-varying annual cycle, and high frequency daily anomalies. For each of these components, mappings are constructed between observations and historical GCM outputs. Future projections are bias-corrected using these mappings, then components are recombined to produce a consistent estimate of future time series. The STAR-ESDM product uses nClimGrid-Daily data with 5-km grid spacing for training over CONUS (Durre et al., 2022). Both the LOCA2 and STAR-ESDM datasets were chosen for operational use in the Fifth National Climate Assessment.

We compare convective and frontal precipitation processes (specifics of how these processes are defined are provided in subsequent paragraphs) from (i) the European Centre for Medium-Range Weather Forecasts Reanalysis fifth Generation data (ERA5; Hersbach

et al., 2020) against (ii) CMIP6 GCMs and their statically downscaled counterparts (LOCA2 and STAR-ESM). We also examine convective and frontal precipitation processes in the observation-based Livneh (Livneh et al. 2015) and nClimGrid-Daily (Durre et al., 2022) hydrometeorological datasets. Finally, to assess the extent to which the stationarity assumption affects projections across statistical and dynamical downscaling, we compare LOCA2 and STAR-ESMD against the

North America component of the Coordinated Regional Downscaling Experiment (NA-CORDEX; Mearns et al. 2017). NA-CORDEX dynamically downscales ERA-Interim reanalysis data and climate model simulations under historical and Representative Concentration Pathway 8.5 W/m$^2$ (RCP8.5) forcings with a suite of regional climate models. We employ five different raw GCM experiments downscaled with five different regional climate models that provide daily

outputs on ~ 25 km resolution. See Table 1 for a summary of all the datasets examined in this paper.

Statistically downscaled products generally only provide a few variables at daily or higher frequencies, which can make it difficult to evaluate covariances. Directly computing covariance between temperature and precipitation at daily timescales may not be useful due to

145 non-linear physical relationships and/or the stochastic nature of weather. We follow Zhang et al. (2023) in isolating for a single convective precipitation event each year by considering precipitation at each grid point coincident with the highest daily maximum temperature during that year (herein convective precipitation). Similarly, we isolate for a single (cold) frontal precipitation event each year by considering precipitation coincident with the greatest drop in

surface temperature for that year (herein convective precipitation). For every grid point, our

method thus identifies one convective precipitation event and one frontal precipitation event per year. To evaluate our method of identifying precipitation events, we (i) identify grid-by-grid the calendar day of convective and frontal precipitation, respectively, for each year over 1980-2014; (ii) create histograms of the number of times that the day of convective or frontal precipitation falls between day 0 and day 365 of each year (days 0 – 365 are thus effectively histogram bins); and (iii) fit a discrete Fourier transform onto the respective histogram to identify the dominant frequency (*i.e.,* frequency corresponding to peak day) present in the data.

We examine daily near-surface temperature and precipitation fields on a per-grid basis during convective and frontal precipitation events over CONUS, focusing on a 21-day window from 10 days prior to and 10 days following the day of convective and frontal precipitation, respectively (and including the day of convective or frontal precipitation itself). For the raw GCMs and ERA5, we also examine moist static energy, which we estimate using daily temperature, specific humidity, and geopotential height, but monthly surface pressure (due to data availability). For the purposes of this study, we examine (i) a 35-year period spanning the 1980-2014 historical interval and (ii) a 35-year period spanning the 2065-2099 interval under the Shared Socioeconomic Pathway "Fossil Fueled Development" scenario with 8.5 W/m$^2$ of radiative forcing (SSP585). For dynamical downscaling outputs, we examine the 2065-2098 interval under the RCP8.5 forcing (note that the years 2006-2014 fall under the RCP8.5 scenario for NA-CORDEX).

## 3. Results

*3.1 Convective and frontal precipitation processes in observation-based datasets*

We first examine convective precipitation in the ERA5 Reanalysis dataset (Figure 1).

Composite time series centered around the hottest day (day 0) shows surface temperature

anomalies increase exponentially from -1K 10 days prior (day -10), peak at 3-4K on the hottest

day (day 0), then decrease exponentially to -1K 10 days following (day + 10). Spatial composites

of the hottest day show warm temperature anomalies over the CONUS domain, while the 5th day

after shows broad cool anomalies. Coincident composite time series of precipitation show

anomalies that decrease from day -10 to day 0 (co-occurring with temperature anomalies

increasing). Precipitation anomalies are lowest on the hottest day (between -1 and -1.5 mm/day),

with the spatial composite of day 0 showing broad dryness. Precipitation anomalies increase

rapidly in the immediate days following, coincident with rapid surface temperature anomaly

decreases. The spatial composite of precipitation on day +5, for instance, shows broad wetting

indicative of convective precipitation.

The above co-evolution of surface temperature and precipitation are consistent with

expectations of convective precipitation: surface temperature will rise until it convects, triggering

precipitation and cooling surface temperature. Analysis of coincident moist static energy

reinforces this mechanism: moist static energy increases until the precipitation event and rapidly

decreases immediately afterwards as the atmosphere stabilizes (Figure 1). Finally, our findings

extend to the observation-based Livneh (Supplemental Figure 1) and nClimGrid-Daily

(Supplemental Figure 2) datasets. Although observational climate datasets themselves have

inherent uncertainties (such as from generation, sampling, or resolution; Zumwald et al. 2020),

strong consistency across ERA5 and the two observation-based products indicate our ERA5

results to be robust.

We next examine cold frontal precipitation in ERA5, centered around the greatest drop in

surface temperature (Figure 2). Our selection of frontal precipitation events show a very different

relationship between temperature and precipitation as compared to convective precipitation.

Composite time series show temperature anomalies to be highest on the day of frontal

precipitation (day 0), drop to the lowest in the two days following (day +1 to +2), then return to

~0 by day +10. Spatial composites of surface temperature show warm anomalies over on day 0

and cold anomalies on day +2. Coincident precipitation time series show anomalies that increase

dramatically (from < 0 mm/day at day -2 to ~4 mm/day at day 0), before falling back to < 0

mm/day. Spatial composites of precipitation anomalies on the day 0 show broad wetting, with

the eastern half of CONUS showing greater anomalies; spatial composites on day +2 show

largely neutral conditions over most of CONUS. Analysis of moist static energy reinforces a cold

frontal precipitation mechanism, with a steep decline in moist static energy that is coincident

with a steep decline in surface temperature and with sudden precipitation.

To further evaluate our method of identifying precipitation events, we apply a discrete

Fourier transform on days of the year when the convective and frontal precipitation events are

occurring, respectively (Figure 3). We find that convective precipitation occurs predominantly in

boreal summer (June-July-August; consistent with when warm days are prevalent). Frontal

precipitation occurs predominantly in boreal winter (December-January-February; consistent

with when cold fronts would be most prevalent), notwithstanding intermountain regions of the

US West where orographic lifting is prevalent (note that this is also the case with raw CMIP6

GCMs; Supplemental Figure 3). We also calculate kernel density estimates (KDE) of

precipitation anomalies before convection (day -2) and after convection (day +2) for the 35-year

composite of convective precipitation events (Figure 4); the two KDEs are significantly different

(p<0.01; Kolmogorov-Smirov test). Moreover, we find 97% of the CONUS grid points to higher

precipitation anomalies at day +2 relative to day -2. We perform similar analyses for frontal

precipitation: KDEs of precipitation anomalies during day +0 and day +1 are significantly

different (p<0.01) from the rest of the 21-day window; 93% of the maximum precipitation in our 35-year composite of events occurs on day +0 or day +1. Given the above-mentioned co-variances, demonstrated skill in selecting for desired events, and expected seasonal occurrence of said events, we deem the physical relationships between surface temperature and precipitation observed in ERA5 during convective and frontal precipitation (as identified in our methodology) to be appropriate for evaluate the credibility of GCMs and their statistically downscaled products.

*3.2 Precipitation processes in raw and statistically downscaled GCMs over the historical interval*

Some spread amongst the GCMs notwithstanding, the eight CMIP6 GCMs herein analyzed behave consistently with ERA5 for both convective and frontal precipitation over the 1980-2014 historical interval (Figure 5; see Table 1 for list of models). That is, convective precipitation anomalies consistently (i) decrease leading up to the hottest day; (ii) are lowest about the hottest day; then (iii) drastically increase with convection in the immediate days following. Frontal precipitation is also clearly visible in the GCMs, with drastic and acute precipitation evident centered around the day of greatest temperature decrease. The raw GCMs not only match the temporal co-evolution of surface temperature and precipitation as demonstrated in ERA5 but correctly simulate the magnitude of anomalies during convective and frontal precipitation events. Our results therefore indicate that the CMIP6 GCMs robustly capture convective and frontal precipitation processes.

We next examine these same co-evolutions after the GCMs are statically downscaled using LOCA2 and STAR-ESDM techniques (Figures 6, 7; note that the same eight models are

examined across the raw GCMs and statistically downscaled data). For temperature, differences

amongst the eight GCMs (*i.e.,* inter-model spread) are noticeably reduced for both convective

and frontal precipitation (see surface temperature time series of the 21 days examined in Figures

6 and 7). This is somewhat expected, as the downscaling method bias-corrects the GCMs to

"match" observations; deviations relative to observations (Livneh dataset for LOCA2 and

nClimGrid-Daily dataset for STAR-ESDM) will thus be minimized. Spatial composites of

downscaled surface temperature, for instance, closely mirror those shown in ERA5 for both

convective and frontal precipitation. Inter-model spread for precipitation can also be reduced,

though this influence is less pronounced than for temperature. Note that bias-correction during

statistical downscaling is performed variable by variable (*i.e.,* independently and without explicit

consideration of local co-variances across variables) and that our definitions of convective and

frontal precipitation in effect selects precipitation fields based on surface temperature

characteristics. Inter-model spread for downscaled precipitation fields are thus not explicitly

prescribed for reduction. Our results suggest that the LOCA2 and STAR-ESDM downscaling

techniques generally preserve co-variances shown in the raw GCMs with high fidelity (compare,

for instance, mean absolute error values for raw GCMs against their downscaled counterparts in

Figures 5-7).

        There are nevertheless clear ensemble-mean differences between the downscaled

products and the raw GCMs (and by extension ERA5 which the raw GCMs simulate with high

skill) that require careful attention. LOCA2 appears nearly identical to ERA5 for convective

precipitation; however, it dampens frontal precipitation relative to ERA5 (and the raw GCMs) by

up to ~2 mm/day (Figures 5, 6). Composite time series show LOCA2 frontal precipitation to

peak at lower anomaly values (2.5 mm/day for the LOCA2 ensemble mean verses 3.9 mm/day in

ERA5); the wet pattern apparent in the ERA5 composite is also diminished in the ensemble

mean spatial composite. Importantly, such dampening is robust across most of the LOCA2 ensemble (Figure 6), indicating it to be an emergent feature of the LOCA2 downscaling method. STAR-ESDM does not exhibit this dampening: it shows frontal precipitation anomalies that closely match the frontal precipitation anomalies of ERA5 and the raw GCMs (Figure 7j-l). STAR-ESDM may slightly overshoot drying anomalies prior to convective precipitation (by less than ~0.5 K); this influence is nevertheless modest and the STAR-ESDM ensemble simulates a range that encapsulates the evolution of frontal precipitation shown in ERA5.

*3.3 Precipitation processes in raw and statistically downscaled GCMs over the future interval*

We next examine convective and frontal precipitation in the raw GCMs over the future interval (2065-2099; Figure 8). For convective precipitation, the co-evolution of surface temperature and precipitation (including the magnitude of their respective anomalies) does not change substantially across the ensemble mean relative to the historical interval (compare Figure 5a-c to Figure 8a-c). For frontal precipitation, however, there is robust intensification that is present across all ensemble members: frontal precipitation peaks at ~ 4 mm/day over the historical interval (Figure 5e) but ~5-6 mm/day in the future interval (Figure 8e). Scatterplots of surface temperature and peak frontal precipitation (Supplemental Figure 4) show steeper associations between the two in the future interval, indicating that frontal precipitation is driven at least in part by temperature increases. Moist static energy levels prior to frontal precipitation are also greater in the future interval relative to the historical interval (compare Figure 5f to Figure 8f), consistent with frontal precipitation intensification.

We again examine these same co-variances after the GCMs are statically downscaled for the future interval. Future interval time series and spatial composite results for both LOCA2 and

STAR-ESDM products appear nearly identical to those of the historical interval, respectively, for convective precipitation (Figures 9a-f, 10a-f). This is consistent with expectations, as the raw GCMs themselves do not show appreciable changes for convective precipitation relative to the historical interval. The robust intensification of frontal precipitation (relative to the historical

interval) simulated by the raw GCMs is not evident in LOCA2 (Figure 9g-i), some slight wetting notwithstanding. LOCA2 dampens frontal precipitation over both the historical and future intervals; the net effect is that it substantially underestimates future frontal precipitation relative to the raw GCMs. For instance, frontal precipitation anomalies reach ~7 mm/day in the raw GCMs but less than 4 mm/day in LOCA2 (and as low as a little as 2 mm/day). Frontal

precipitation is intensified in STAR-ESDM (Figure 10j-l; ~4-6 mm/day in the future interval compared to ~3-5 mm/day in the historical interval), although the magnitude of the intensification falls short of what is simulated by the raw GCMs.

*3.4 Precipitation processes in NA-CORDEX dynamical downscaling*

Finally, we examine how convective and frontal precipitation processes are affected post *dynamical* downscaling across five different regional climate models. Dynamical downscaling of ERA-Interim preserves expected hydroclimate covariances during convective and frontal precipitation processes (Supplemental Figure 5; note that inter-model differences are entirely attributable to regional climate models as the underlying data being downscaled is identical

across the five models). Biases in regional climate models appear to be relatively small and are not prohibitive in representing convective and frontal precipitation processes on local-scales. These biases are also small when GCM data, instead of observation data, is downscaled. Convective precipitation processes in dynamical downscaled GCM data in the future interval do

not change much relative to the historical interval, consistent with the raw GCMs and with

315 statistical downscaling (Figure 11). However, we find that dynamical downscaling preserves

robust intensification of future frontal precipitation simulated by raw GCMs, in strong contrast to

the dampening of this intensification seen with statistical downscaling (Figure 12). For instance,

frontal precipitation in the future interval of dynamically downscaled GCM data is ~1.5 mm/day

to 2 mm/day greater than dynamically downscaled GCM data in the historical interval, consistent

with the magnitude of intensification seen with the raw GCMs (Figure 12, Supplemental Figure

6). This finding is robust across all five regional climate models examined, indicating low

sensitivity to regional model biases.

## 4. Conclusions

Using (only) surface temperature and precipitation outputs, we have employed

convective and frontal precipitation mechanisms to evaluate the credibility of statistical (and

dynamical) downscaling products. We find that the LOCA2 and STAR-ESDM statistical

downscaling techniques generally preserve expected covariances between temperature and

precipitation during convective precipitation over both the historical and future intervals.

Statistical downscaling also preserves expected covariances of temperature and precipitation

during frontal precipitation events over the historical interval; however, it dampens projected

intensification of frontal precipitation in the future interval that is otherwise robustly simulated in

the raw CMIP6 GCMs (*i.e.,* prior to downscaling) and with dynamical downscaling.

Convective precipitation in the raw GCMs as examined in our analyses does not exhibit

material differences across the historical and future intervals (as opposed to frontal precipitation

which shows robust intensification in the future interval). Convective precipitation is therefore

likely more insensitive to the stationarity assumption, notwithstanding the possibility that CMIP6 models themselves may not effectively resolve global cloud-systems (and thus may not capture non-stationary changes in convective precipitation). Frontal precipitation, on the other hand,

shows robust intensification over the future interval, providing a useful evaluation case into the (in)ability of historical functional relationships inherent to statistical downscaling to resolve non-stationary phenomena. Indeed, the dampening of frontal precipitation shown suggests that LOCA2 and STAR-ESDM may not appropriately capture structural changes to meteorological phenomena. This is in strong contrast to dynamical downscaling (regardless of the regional

climate model chosen), which preserves non-stationary physical relationships among variables.

Our results are, to some extent, qualitatively intuitive: common statistical downscaling methods apply historical functional relationships to the future under the assumption that they will be preserved despite climate change. It is therefore somewhat expected that such techniques will may underestimate changes within non-stationary phenomena. This effect should be

acknowledged when estimating the magnitude of future change, particularly when considering the dominant (*e.g.,* Baek et al. 2019, 2021) and/or non-stationary (*e.g.,* Baek et al. 2020; Scholz et al. 2022) nature of internal atmospheric variability in driving hydrologic hazards. Evaluation frameworks clearly demonstrating this to be the case have nevertheless proved elusive. Our work addresses this important gap by demonstrating divergence between statistically and dynamically

downscaled methods when estimating enhancement of frontal precipitation (an example of non-stationary process testable with just daily surface temperature and precipitation). These same issues are likely to arise among data-driven (i.e., machine learning based) climate models, particularly if those methods are only trained on historical data and subsequently used for future projections. Equally importantly, our work highlights expected co-evolution of precipitation and

temperature during convective and frontal precipitation events as process-based evaluation

diagnostics that can be applied universally to a wide range of statistically downscaled products.

**Tables**

| (a) CMIP6 Models Analyzed | |
|---|---|
| CCCma CanESM5 (2.8°x2.8°) | INM-CM5-0 (2.0°x1.5°) |
| AS-RCEC TaiESM1* (1.25°x0.9°) | NCC NorESM2-MM (1.25°x0.9°) |
| CAS FGOALS-g3 (2.0°x5.2°) | NOAA GFDL-ESM4 (1.25°x1.0°) |
| Earth-Consortium EC-Earth3 (0.7°x0.7°) | BCC BCC-CSM2-MR (1.125°x1.1°) |
| **(b) Observation-based Datasets Analyzed** | |
| ERA5 (0.25°x0.25°) | |
| Livneh (6-km grid) | |
| nClimGrid-Daily (5-km grid) | |
| **(c) Statistically Downscaled Data Analyzed** | |
| LOCA2 (6-km grid) | |
| STAR-ESDM (5-km grid) | |

| (d) NA-CORDEX (Dynamically Downscaled) Data Analyzed | | | | | |
|---|---|---|---|---|---|
| RCM \ BC | RegCM4 | WRF | CRCM5-OUR | CRCM5-UQAM | CanRCM4 |
| ERA-Int | 25 km | 25 km | 0.22° | 0.22° | 0.22° |
| MOHC HadGEM2-ES | 25 km | | | | |
| CCCma CanESM2 | | | | | 0.22° |
| MPI-M MPI-ESM-MR | | | | 0.22° | |
| CERFACS CNRM-CM5 | | | 0.22° | | |
| NOAA GFDL-ESM2M | | 25 km | | | |

**Table 1: (a)** List of CMIP6 models analyzed. All models use the r1i1p1f1 member. We examine the same eight models in the LOCA2 and STAR-ESDM downscaled data. *TaiESM1 is only analyzed over the historical interval (and not the future interval) for the raw GCM due to data availability. Longitude by latitude grid resolution is provided in parenthesis (rounded to nearest tenth of a degree except in cases where resolution ends exactly in a quarter or eighth of a degree). **(b)** List of observation-based datasets analyzed. **(c)** List of statistically downscaled datasets analyzed. **(d)** Simulation matrix adapted from NA-CORDEX. Left column shows the underlying boundary condition (BC) data being dynamically downscaled. Top row shows the regional climate model (RCM) driving the downscaling. The simulations analyzed show the grid-spacing of downscaled model. In addition to global climate models, NA-CORDEX downscales ERA-Interim (top of left column) across different regional climate models (this allows for a comparison of downscaling across a common dataset).

**Figures**

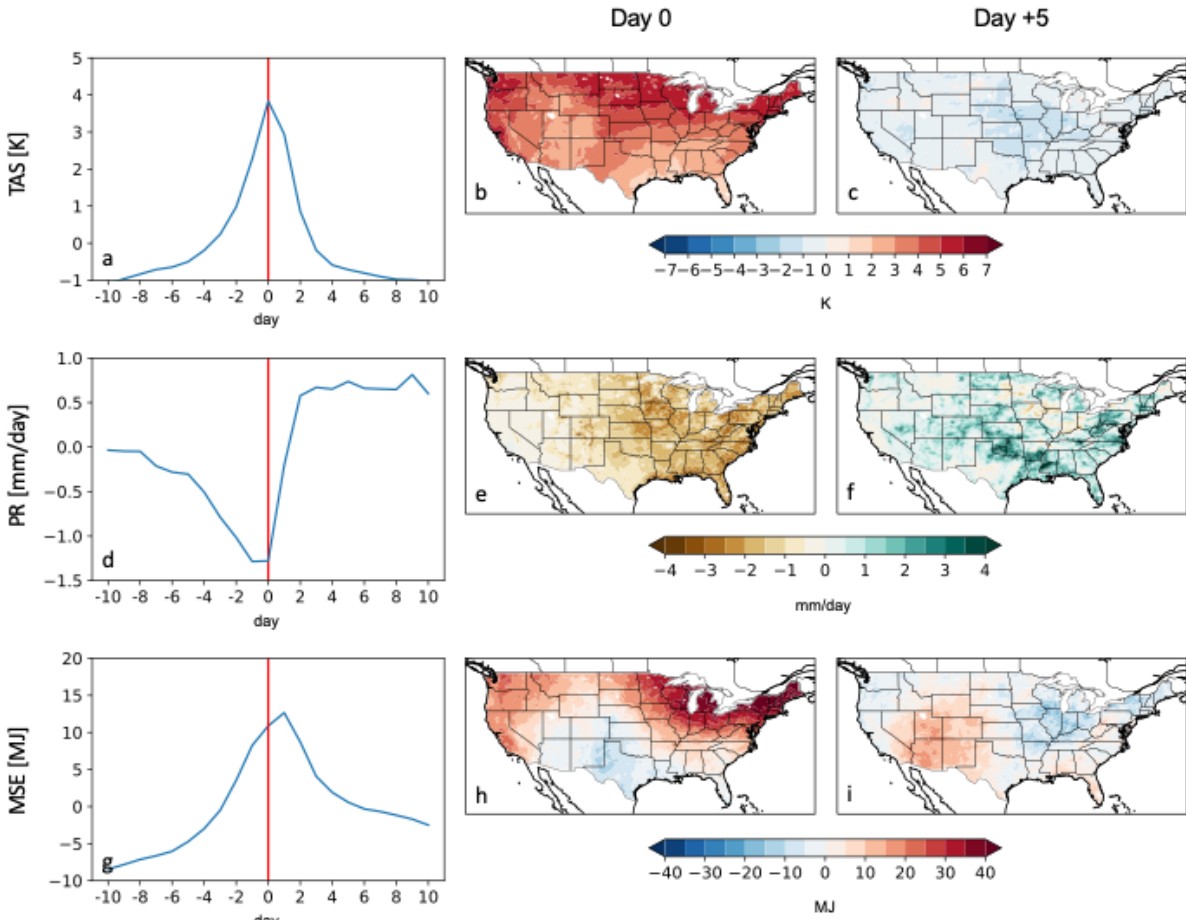

**Figure 1: (a)** 21-day composite (spatially averaged over contiguous US (CONUS) domain) time series of surface temperature anomalies (relative to 21 day average) centered around the day of convective precipitation using ERA5 data over the 1980-2014 interval. **(b)** Spatial composite of surface temperature anomalies on the day of convective precipitation **(c)** Spatial composite of surface temperature anomalies 5 days after convective precipitation **(d-f)** Same as (a-c) but for precipitation. Note that these are anomalies relative to the 21 day window, yielding both positive and negative values. **(g-i)** Same as (a-c) but for moist static energy (MSE).

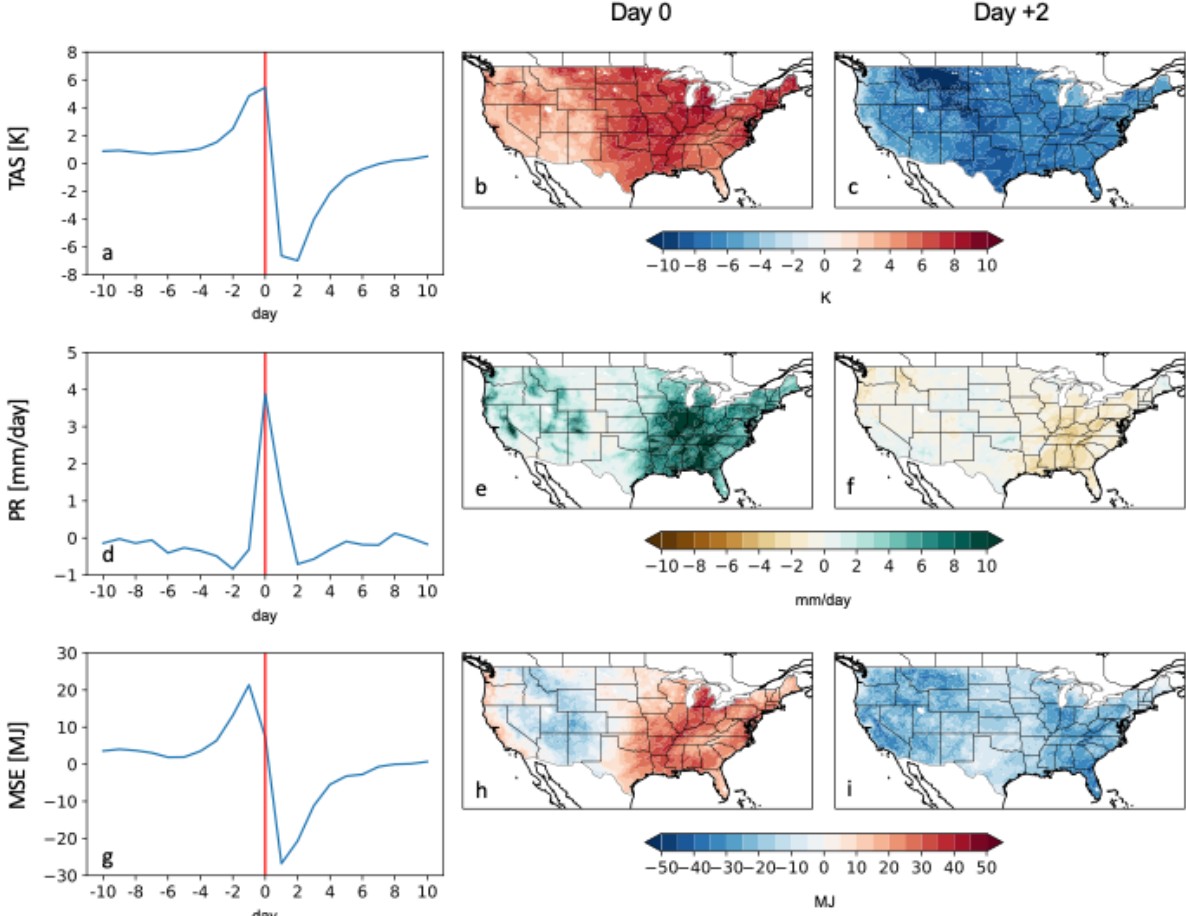

**Figure 2: (a)** 21-day composite (spatially averaged over CONUS domain) time series of surface temperature anomalies (relative to 21 day average) centered around the day of cold frontal precipitation using ERA5 data over the 1980-2014 interval. **(b)** Spatial composite of surface temperature anomalies on the day of convective precipitation **(c)** Spatial composite of surface temperature anomalies two days following the day of frontal precipitation **(d-f)** Same as (a-c) but for precipitation. **(g-i)** Same as (a-c) but for moist static energy.

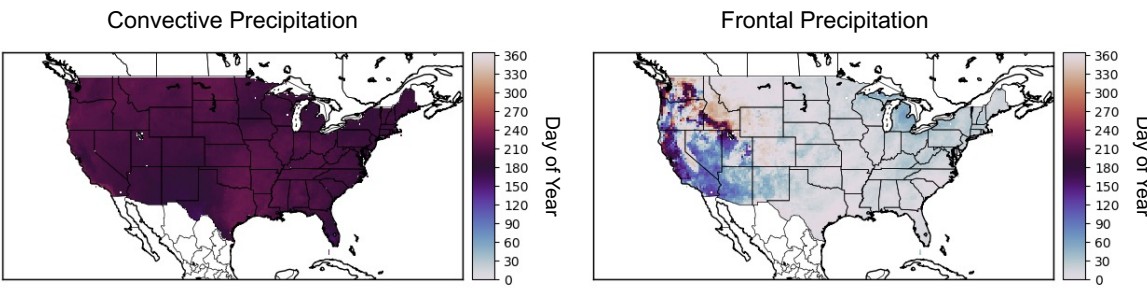

**Figure 3:** Peak convective and frontal day of year using ERA5 dataset. Peak day is determined using a discrete Fourier transform.

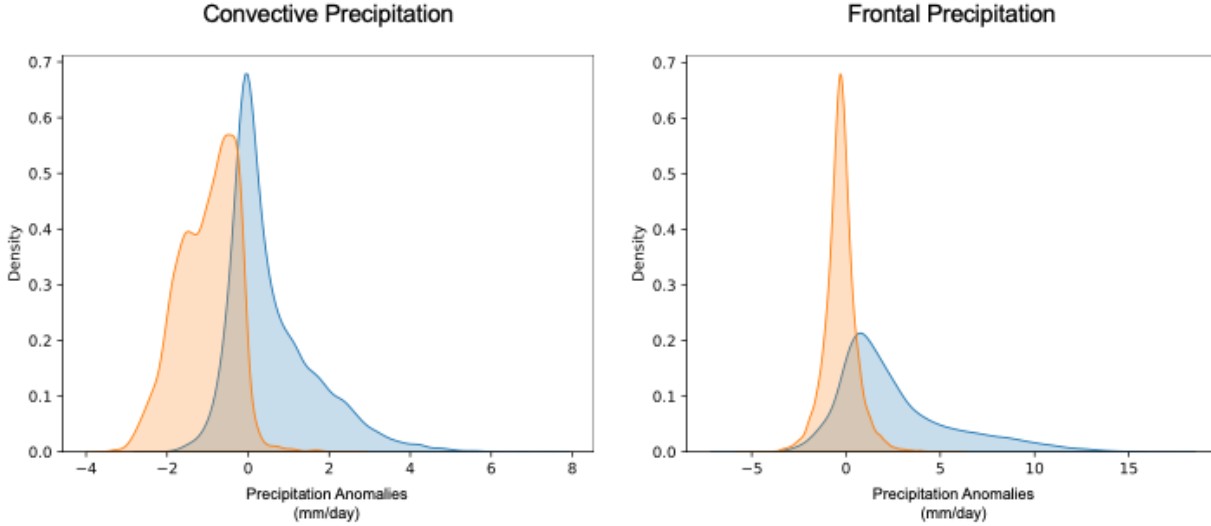

**Figure 4: (left)** Kernel density estimates (KDE) of convective precipitation anomalies before convection (orange; day -2) and after convection (blue; day +2) for the 35-year composite of convective precipitation events. 97% of grid points during the 21-days analyzed show higher precipitation anomalies after convection. The two KDEs are significantly different (p<0.01) as determined by a Kolmogorov-Smirnov test. **(right)** Kernel density estimates of frontal precipitation anomalies on day +0 and day +1 (blue) and all other days of the 21-day window analyzed (orange; randomly sampled). 93% of the maximum precipitation occur on day +0 or day +1. The two KDEs are significantly different (p<0.01) as determined by a Kolmogorov-Smirnov test.

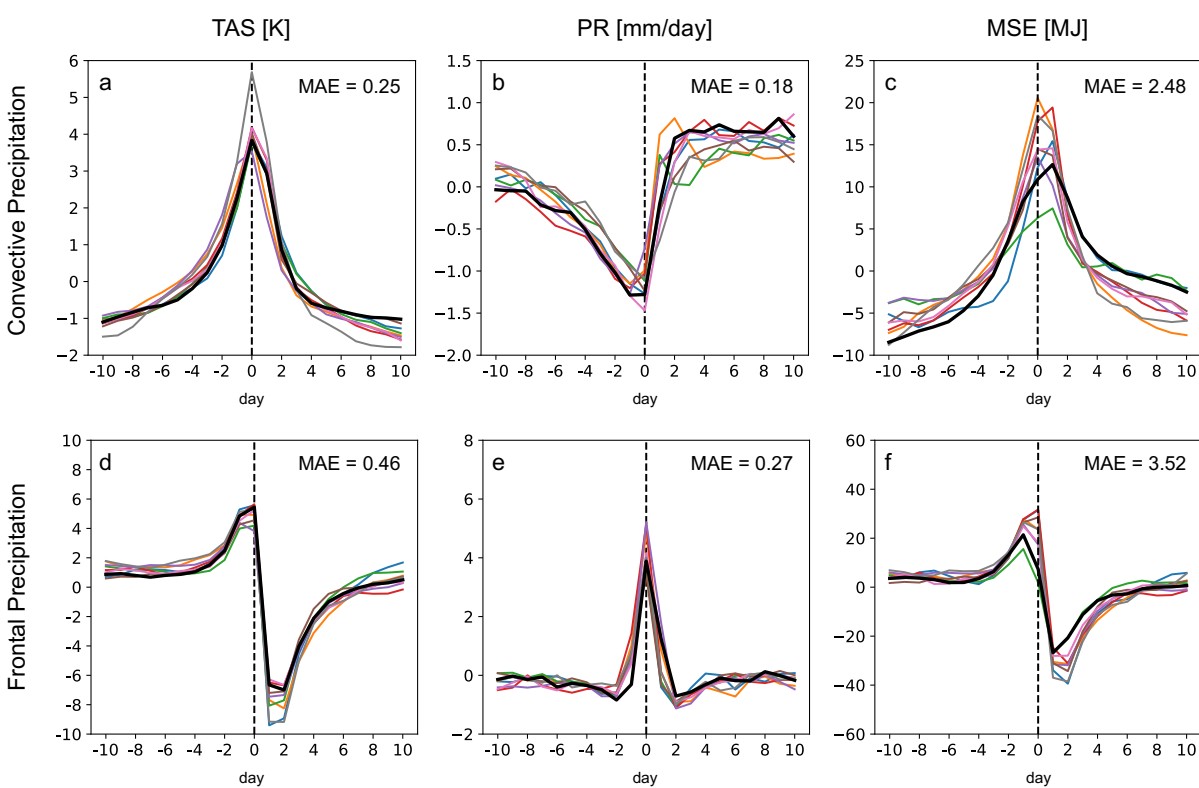

**Figure 5:** 21-day composite time series (spatially averaged over CONUS domain) of **(a)** surface temperature (K), **(b)** precipitation (mm/day), and **(c)** moist static energy ($10^7$ Joules) anomalies (relative to 21 day average) for (colored lines; list of GCMs provided in Table 1) raw CMIP6 GCM and (solid black line) ERA5 data. Time series are centered around the day of convective precipitation and for the 1980-2014 period **(d-f)** Same as (a-c) but for frontal precipitation. Mean absolute error (MAE) is calculated between ERA5 time series and CMIP6 time series and provided in upper right corner of plots.

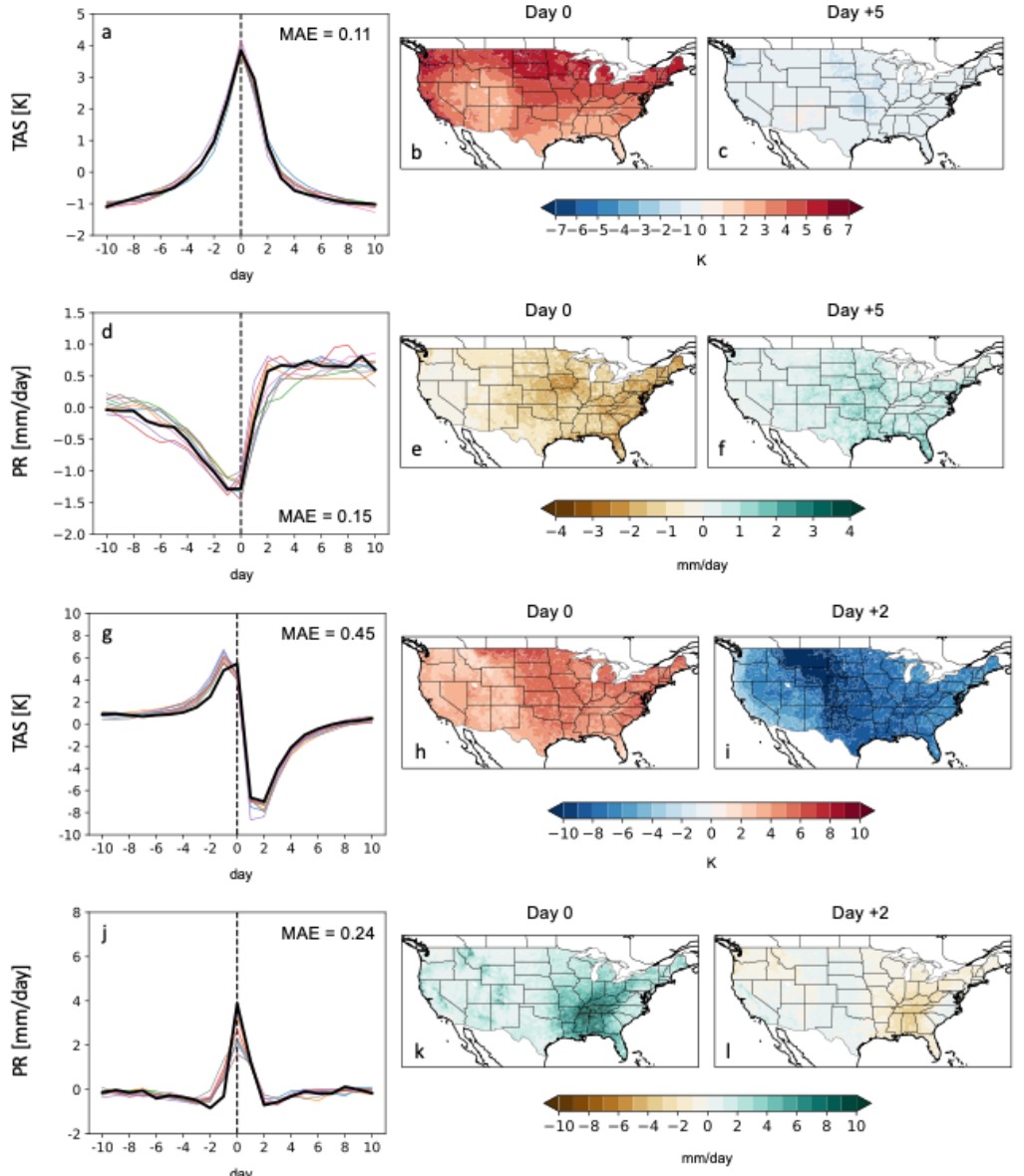

**Figure 6: (a)** 21-day composite (spatially averaged over CONUS domain) time series of surface temperature anomalies (relative to 21 day average) centered around the day of convective precipitation using (colored lines) LOCA2 data over the 1980-2014 interval. Solid black line shows ERA5 data. **(b)** Spatial composite of surface temperature on the day of convective precipitation using LOCA2. **(c)** Spatial composite of surface temperature 10 days prior to convective precipitation using LOCA2 **(d-f)** Same as (a-c) but for precipitation. **(g)** 21-day composite time series (spatially averaged over CONUS domain) of surface temperature anomalies (relative to 21 day average) centered around the day of frontal precipitation using (colored lines) LOCA2 data over the 1980-2014 interval. Solid black line shows ERA5 data. **(h)** Spatial composite of surface temperature on the day of convective precipitation using LOCA2 data. **(i)** Spatial composite of surface temperature anomalies 10 days prior to convective precipitation using LOCA2 data **(j-l)** Same as (g-i) but for precipitation. Mean absolute error (MAE) is calculated between (i) ERA5 time series and (ii) LOCA2 time series and provided in corner of (a, d, g, j).

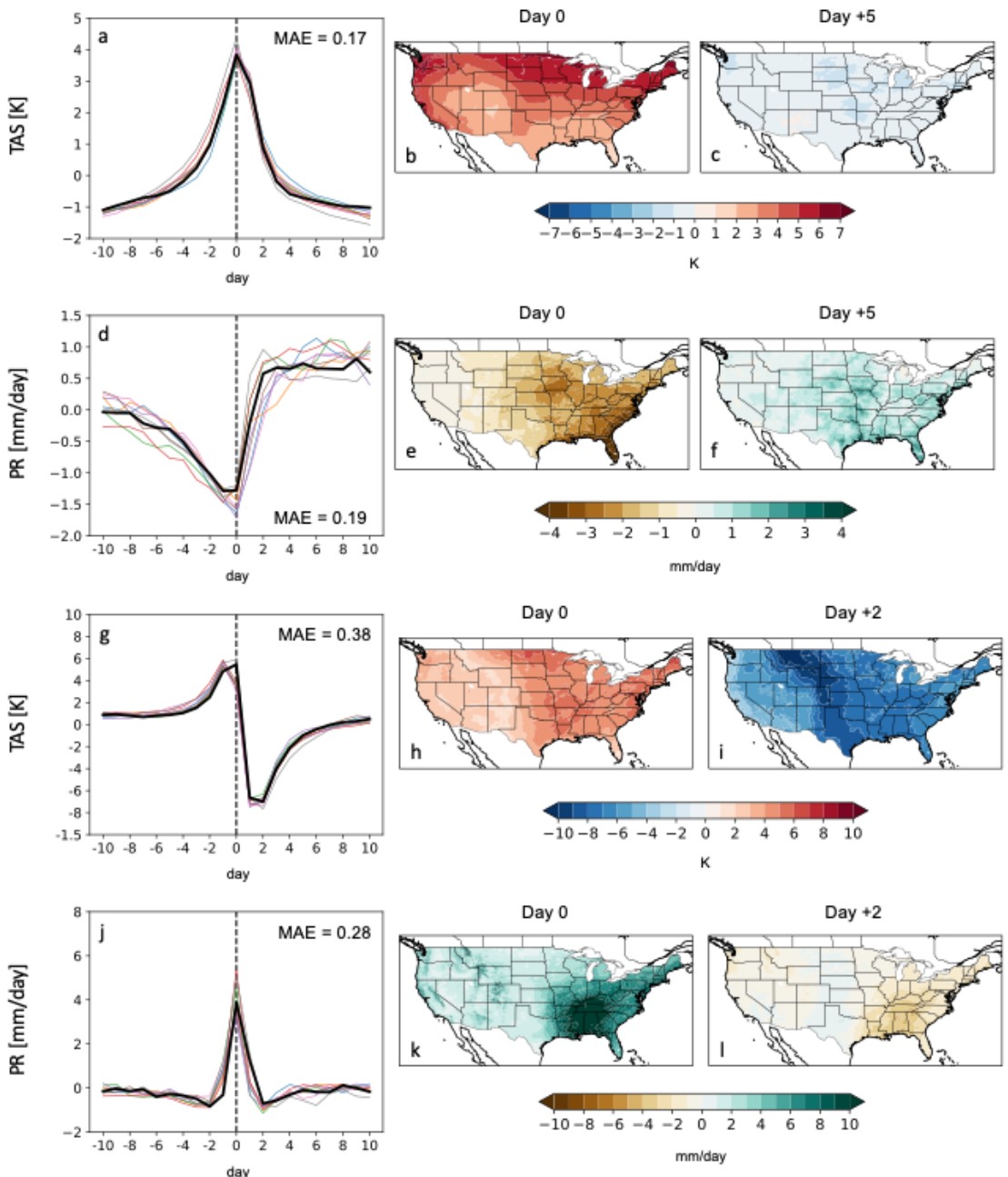

**Figure 7:** Same as Figure 6, but for STAR-ESDM data.

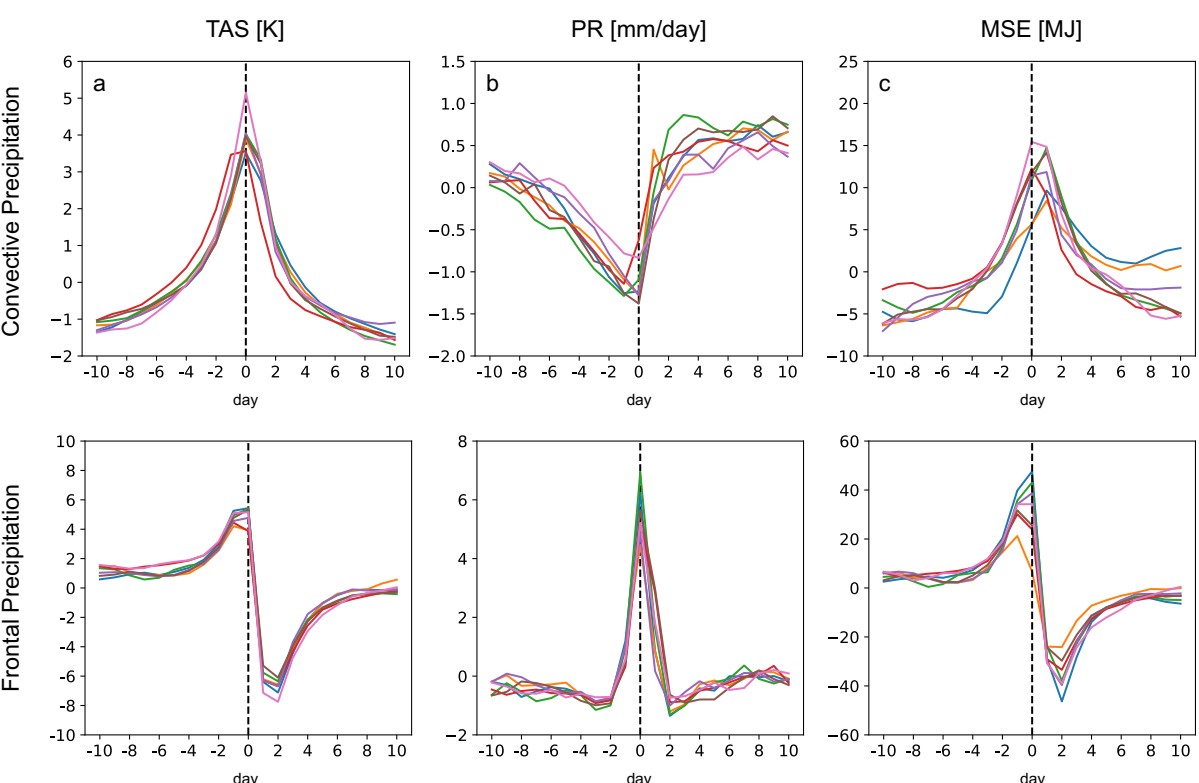

**Figure 8:** Same as Figure 5, but for the 2065-2099 interval under SSP585 forcing.

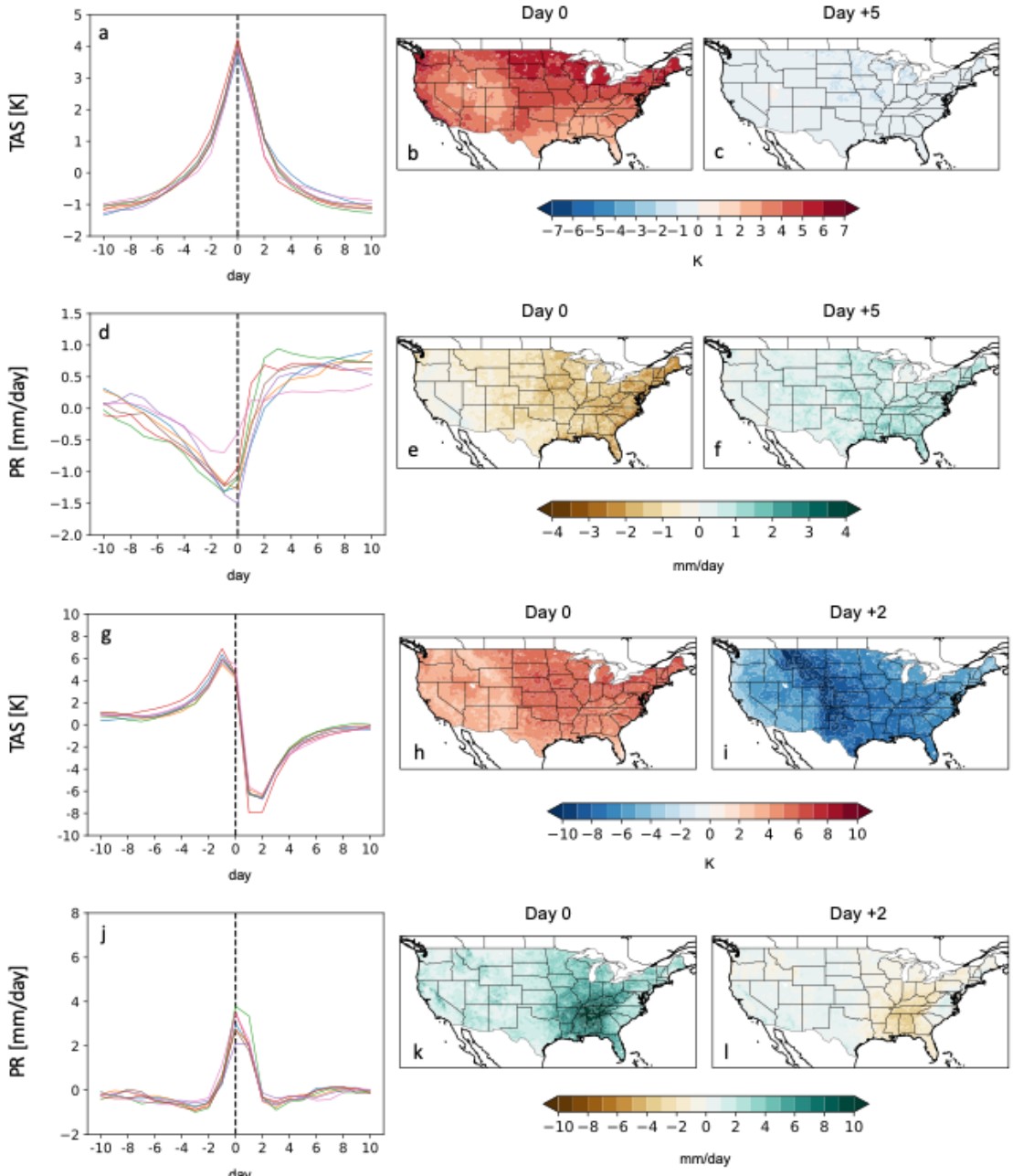

**Figure 9:** Same as Figure 6, but for the 2065-2099 interval under SSP585 forcing.

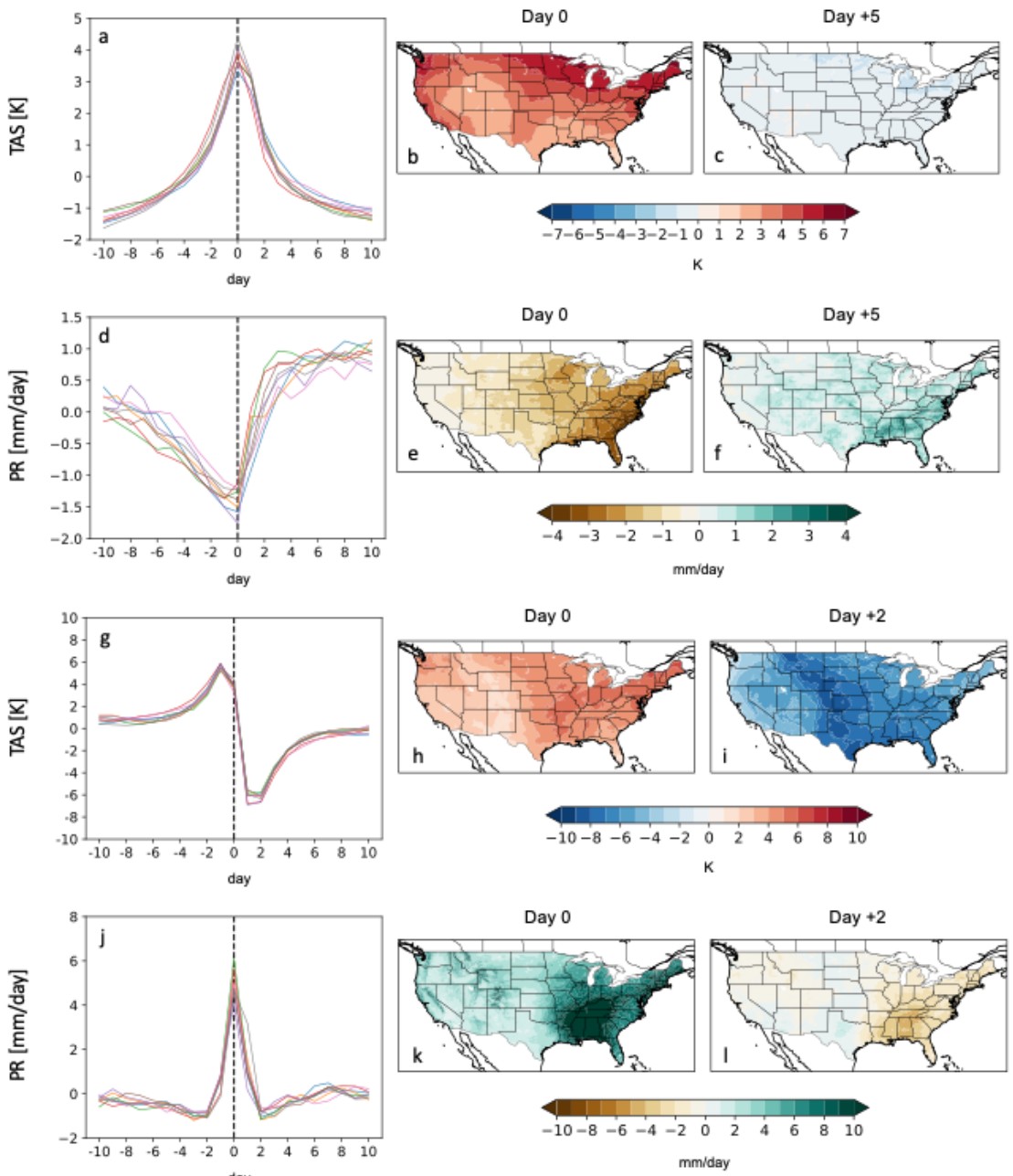

**Figure 10:** Same as Figure 7, but for STAR-ESDM data.

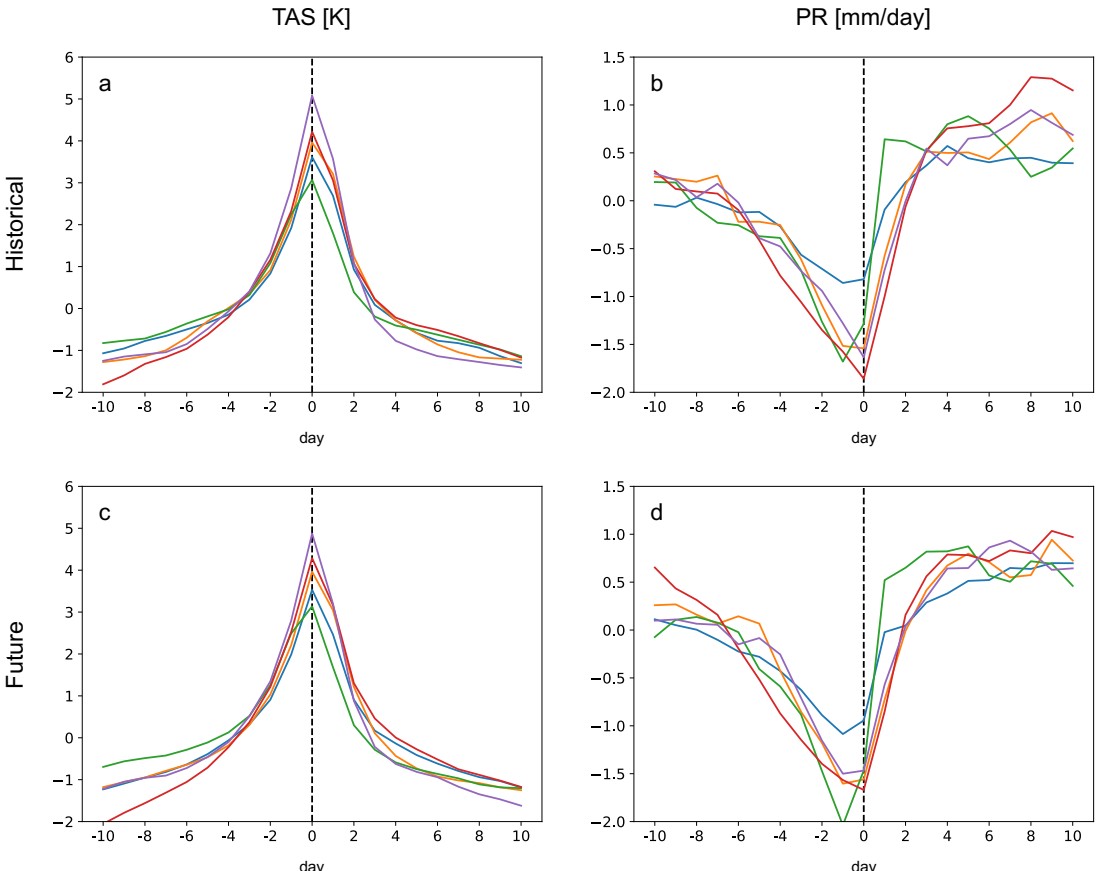

**Figure 11:** 21-day composite time series of CONUS **(a)** surface temperature and **(b)** precipitation anomalies (relative to 21 day average) centered around the day of convective precipitation using dynamical downscaling of ERA-Interim data over the 1989-2009 interval. **(c-d)** Same as (a-b) but for the future interval over 2065-2098 under RCP8.5 forcing.

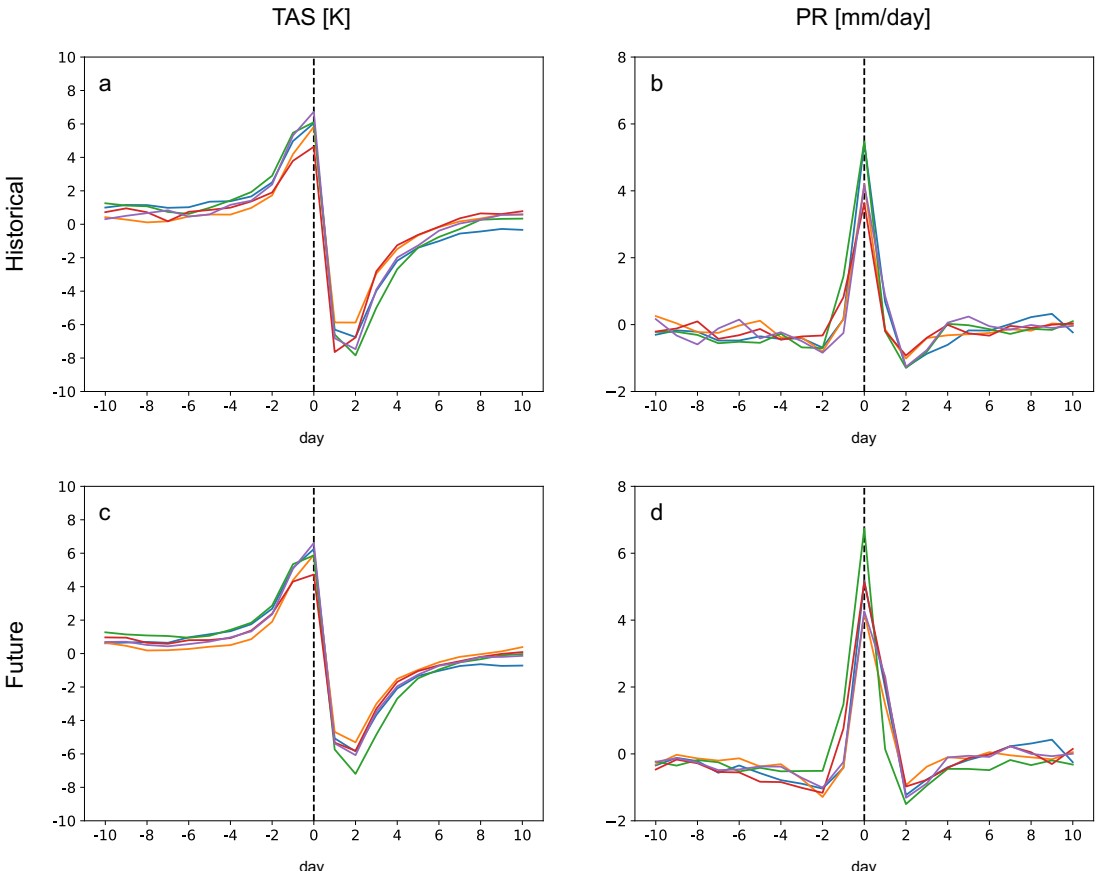

**Figure 12:** 21-day composite (spatially averaged over CONUS domain) time series of **(a)** surface temperature anomalies (relative to 21 day average) and **(b)** precipitation anomalies (relative to 21 day average) centered around the day of convective precipitation using NA-CORDEX dynamical downscaling of GCM data over the 1980-2014 interval. **(c-d)** Same as (a-b) but for the future interval over 2065-2098 under RCP8.5 forcing. Note that the years 2006-2014 fall under the RCP8.5 scenario for NA-CORDEX.

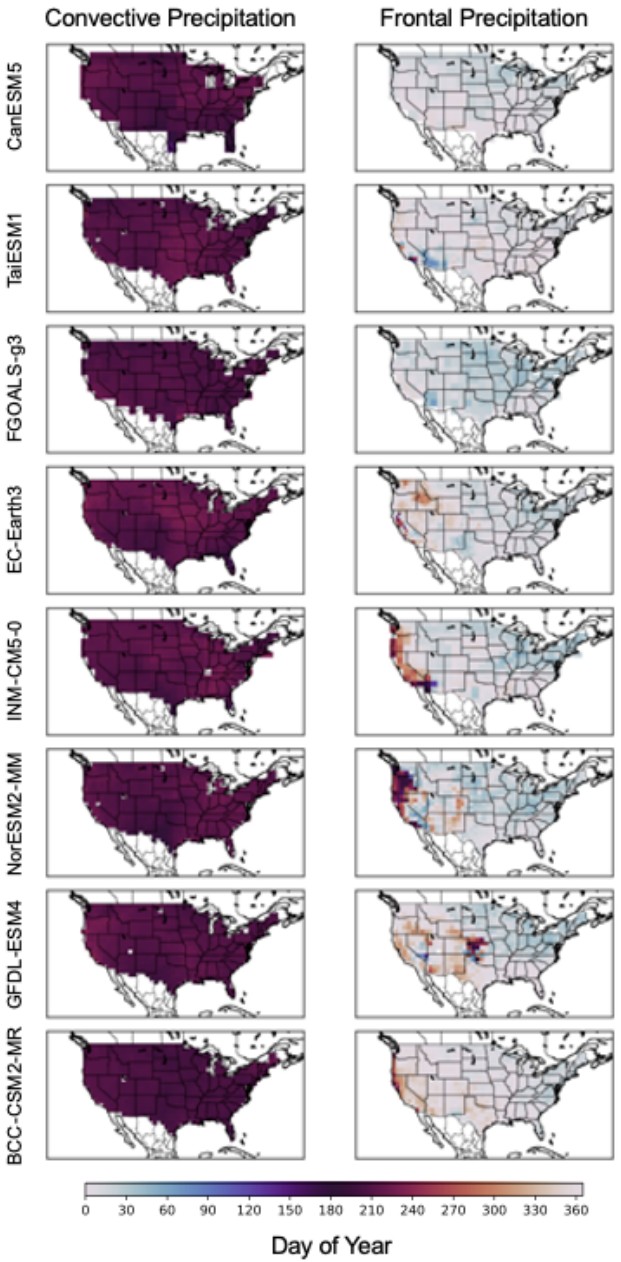

**Supplemental Figure 1**: Same as Figure 3 but for the 8 raw CMIP6 GCMs.

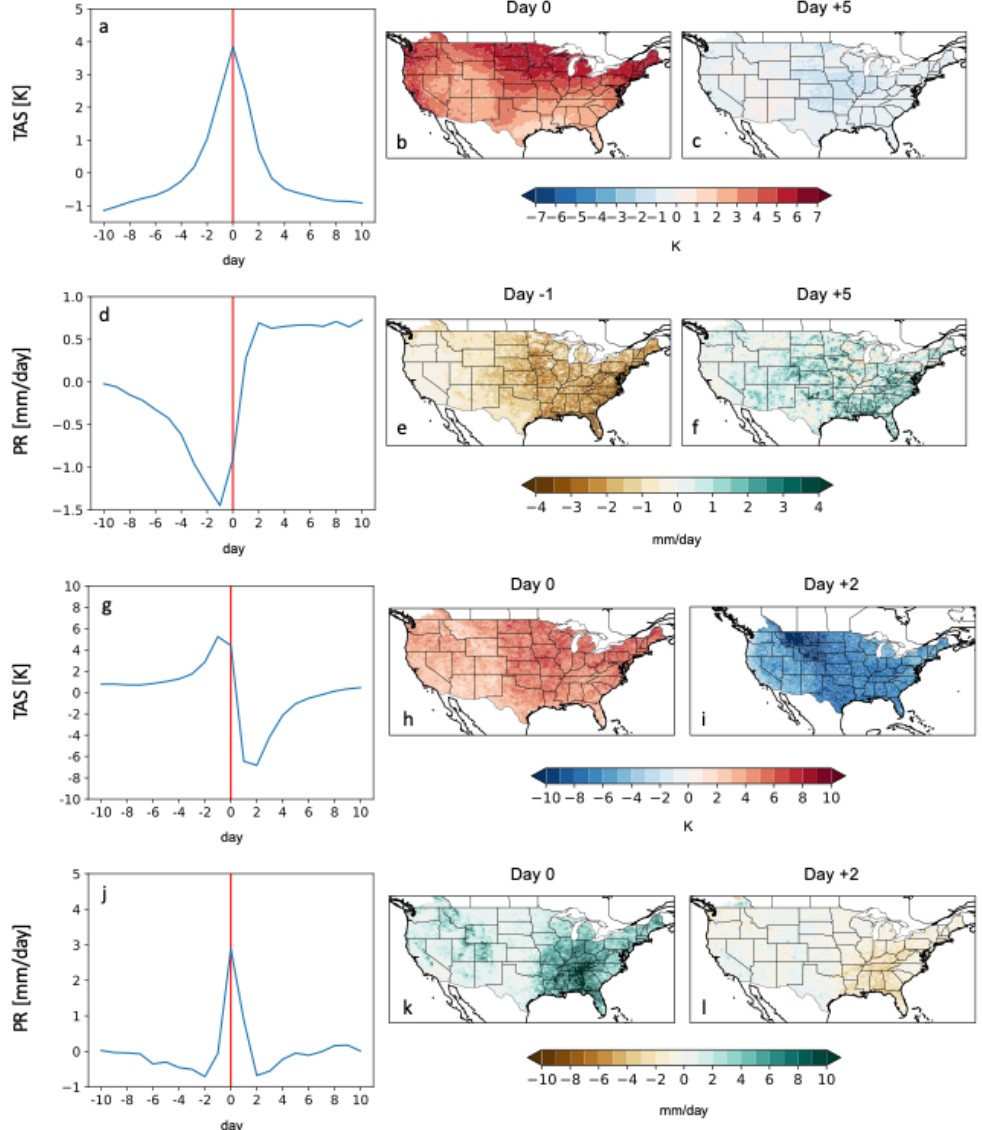

**Supplemental Figure 2: (a)** 21-day composite (spatially averaged over CONUS domain) time series of surface temperature anomalies (relative to 21 day average) centered around the day of convective precipitation using Livneh data over the 1980-2014 interval. **(b)** Spatial composite of surface temperature anomalies on the day of convective precipitation **(c)** Spatial composite of

surface temperature anomalies 5 days after convective precipitation. **(d)** 21-day composite time series of CONUS precipitation anomalies (relative to 21 day average) centered around the day of convective precipitation using Livneh data over the 1980-2014 interval. Note that these are anomalies relative to the 21 day window examined, yielding both positive and negative values. **(e)** Spatial composite of precipitation anomalies on the day prior to convective precipitation **(f)**

Spatial composite of surface temperature anomalies 5 days after convective precipitation. **(g)** 21-day composite time series of CONUS surface temperature anomalies (relative to 21 day average) centered around the day of frontal precipitation using Livneh data over the 1980-2014 interval. **(h)** Spatial composite of surface temperature anomalies on the day of frontal precipitation. **(i)** Spatial composite of surface temperature anomalies two days after convective precipitation. **(j-l)**

Same as (g-i) but for precipitation.

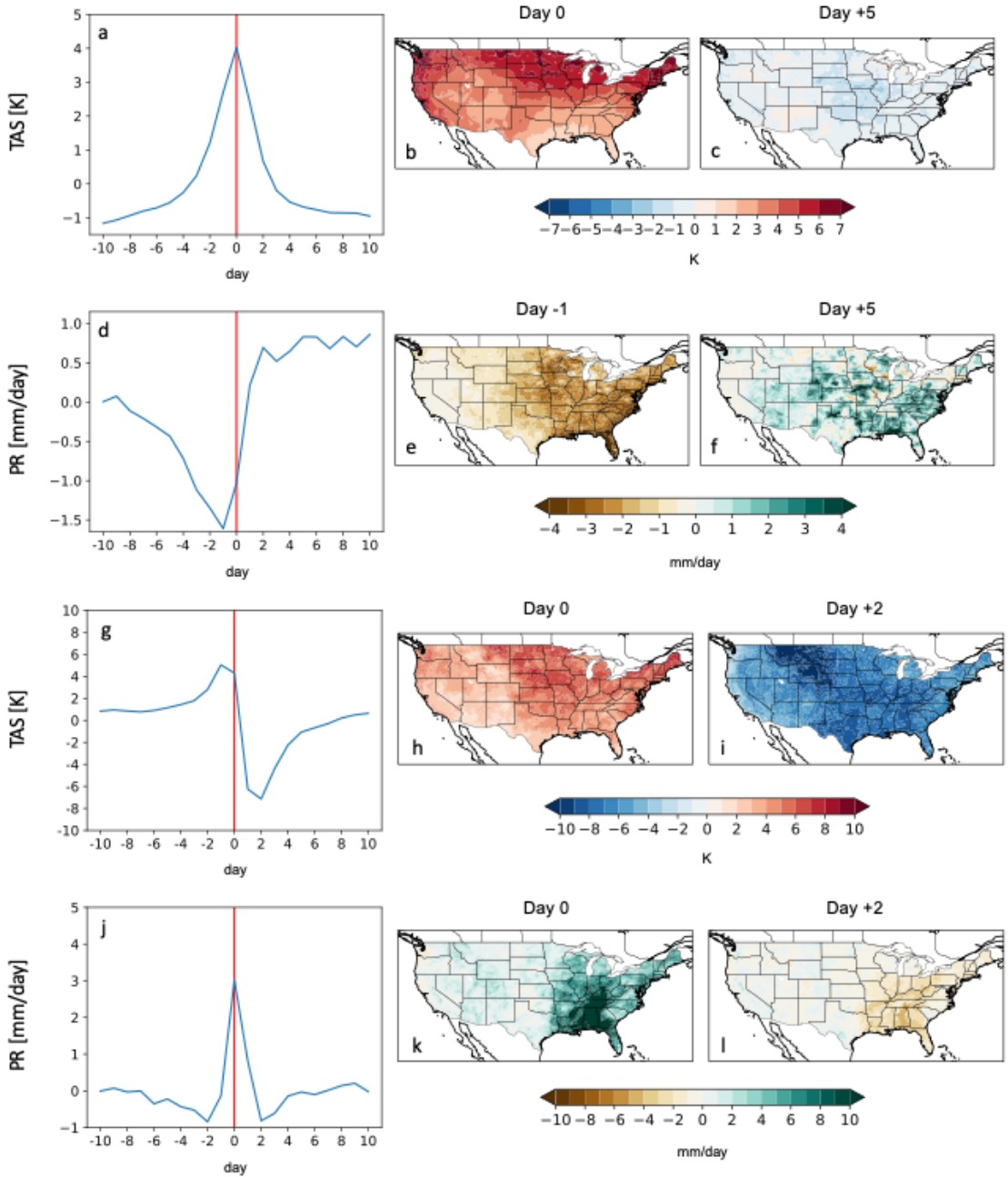

**Supplemental Figure 3:** Same as Supplemental Figure 2 but for the nClimGrid-Daily data over 1980-2005.

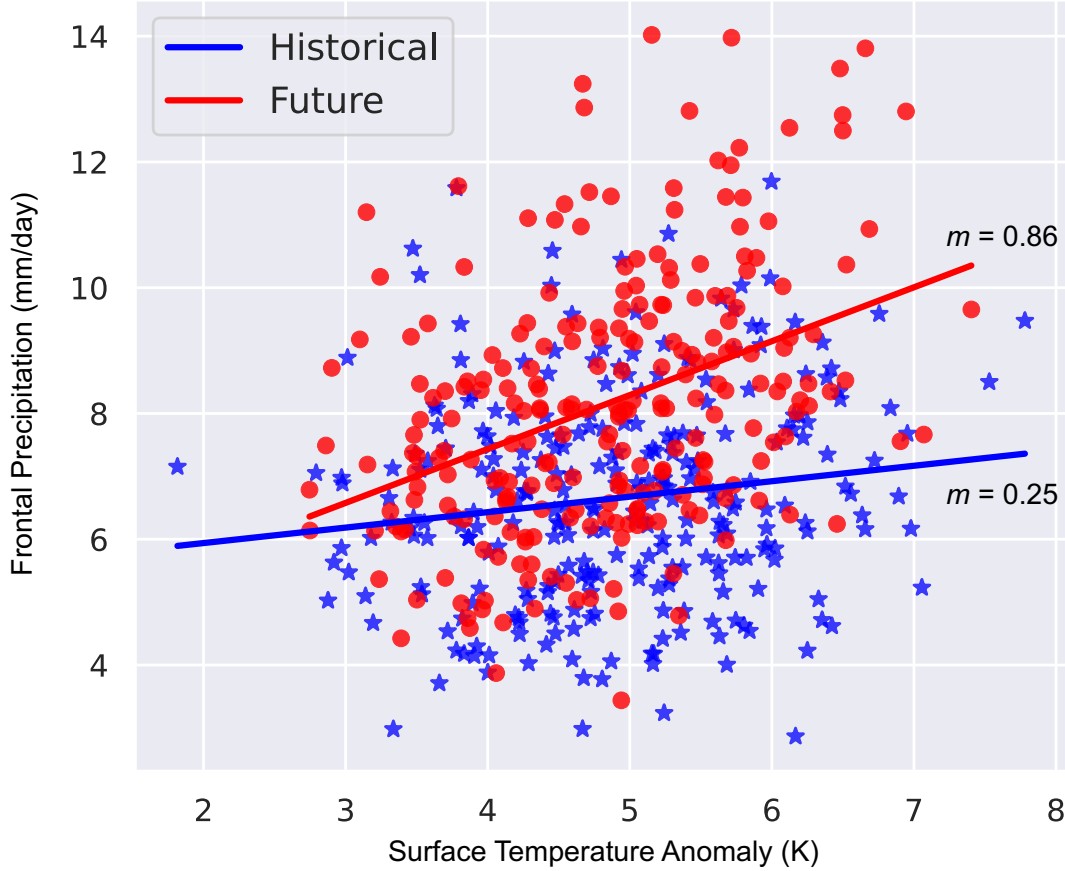

**Supplemental Figure 4:** Scatterplots of surface temperature and frontal precipitation centered
on the day of greatest surface temperature drop for raw CMIP6 models over (red) 1980-2014 and
(blue) 2065-2099. A linear regression model (slope indicated by *m*) is fitted using all the
different models together over the two periods.

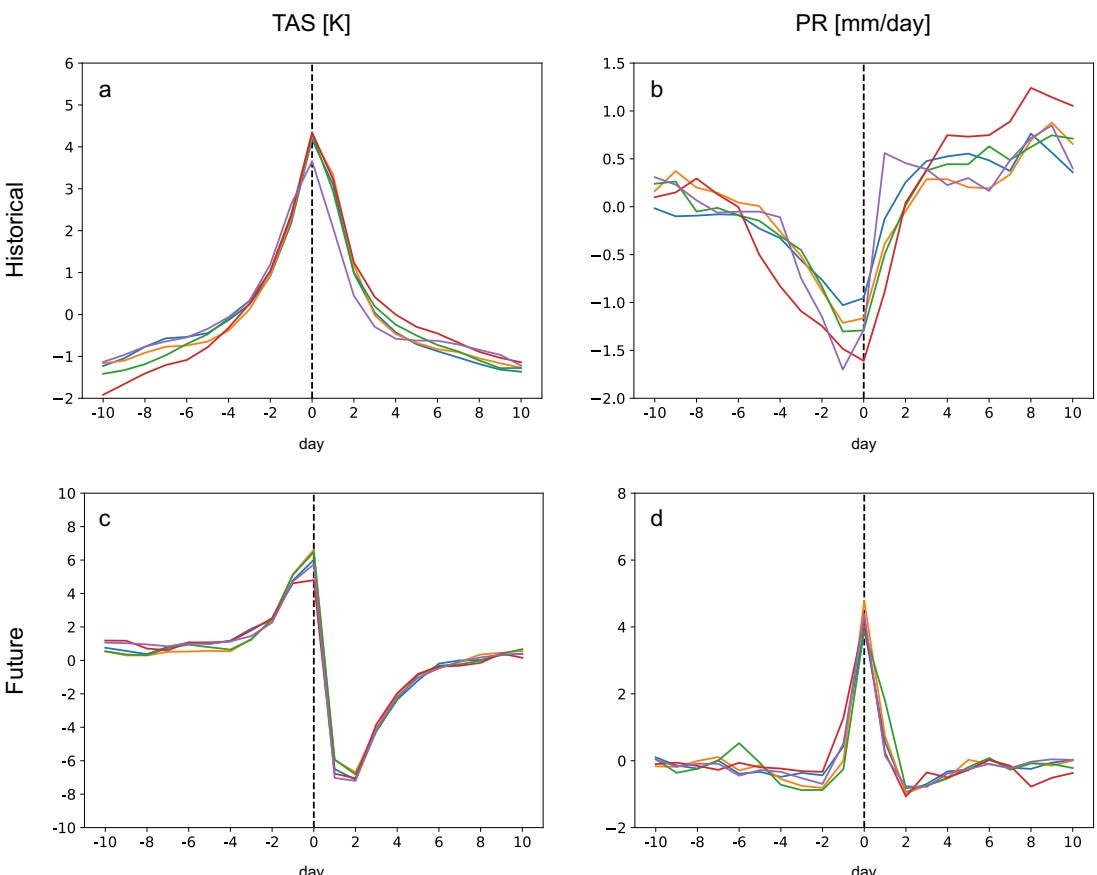

**Supplemental Figure 5:** 21-day composite time series of CONUS **(a)** surface temperature and **(b)** precipitation anomalies (relative to 21 day average) centered around the day of convective precipitation using dynamical downscaling of ERA-Interim data over the 1989-2009 interval. **(c-d)** Same as (a-b) but for frontal precipitation.

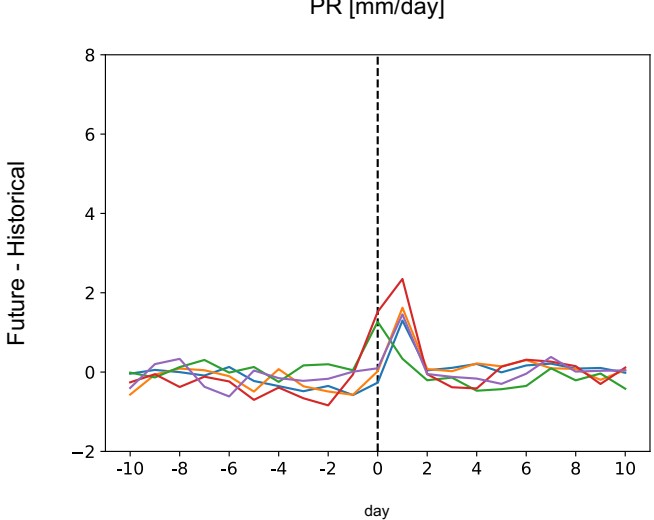

**Supplemental Figure 6:** Difference between Figure 12d and Figure 12b.

**Code and Data Availability**

Code required to conduct the analyses herein are available on
https://zenodo.org/records/11194306. All data used in this study are publicly available. The raw
CMIP6 GCM data can be downloaded from the USA portal of the Earth System Grid Federation
(https://aims2.llnl.gov/search/cmip6/). ERA5 data can be downloaded from the Copernicus
Climate Data Store (https://cds.climate.copernicus.eu/cdsapp#!/dataset/reanalysis-era5-single-levels?tab=form). NA-CORDEX data can be downloaded from the National Center for
Atmospheric Research Climate Data Gateway
(https://www.earthsystemgrid.org/search/cordexsearch.html). Livneh data can be downloaded
from the National Centers for Environmental Information at
(https://www.ncei.noaa.gov/access/metadata/landing-page/bin/iso?id=gov.noaa.nodc:0129374).
The nClimGrid-Daily data can also be downloaded from the National Centers for Environmental
Information at (https://www.ncei.noaa.gov/products/land-based-station/nclimgrid-daily). LOCA2
data can be downloaded from (https://cirrus.ucsd.edu/~pierce/LOCA2/). The STAR-ESDM data
can be downloaded from (https://app.globus.org/file-manager?origin_id=9d6d994a-6d04-11e5-ba46-22000b92c6ec&origin_path=%2Fglobal%2Fcfs%2Fprojectdirs%2Fm3522%2Fcmip6%2FSTAR-ESDM%2Fssp585%2F&two_pane=true).

**Author contributions**

S.H.B. and P.A.U. designed the study. S.H.B. performed the analyses and wrote the paper, with
contributions from all co-authors.

**Competing interests**

One of the authors is a member of the editorial board of Geoscientific Model Development.

**Acknowledgements**

This work was performed under the auspices of the U.S. Department of Energy by Lawrence
Livermore National Laboratory under contract DE-AC52-07NA27344. The authors would like to
acknowledge the Lab Directed Research and Development program 50385-23ERD050. LLNL-JRNL863969.

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
