# Peer review of "Evaluating downscaled products with expected hydroclimatic co-variances"

_EGUsphere, 2024_

## Author Comment (AC1)

RESPONSE TO REVIEWER #2 FOR *GEOSCIENTIFIC MODEL DEVELOPMENT*:
MANUSCRIPT EGUSPHERE-2024-1456
BY SEUNG H. BAEK, PAUL. A. ULLRICH, BO DONG, AND JIWOO LEE

We thank Reviewer #2 for thoughtful and constructive feedback. This Response to the Reviewer file provides a complete documentation of the changes that have been made in response to each individual comment. Reviewer's comments are shown in plain text. Authors' responses are shown in **bold**. Quotations from the revised manuscript are shown in ***bold italics***.

**Reviewer #2**

Thank you for inviting me to review the paper: "Evaluating downscaled products with expected hydroclimatic co-variances" by Baek et al.

First, please accept my apologies for the slowness in returning this review.

This paper is important and interesting because rather than just considering climate variables of concern in isolation (e.g., their role in extremes), the manuscript emphasizes generating the correct covariances between quantities when downscaling. On that basis, I hope the manuscript will eventually appear in print in some format.

The two obvious variables needing consistency are temperature and heavy precipitation, as noted in the Abstract.

Below are some suggestions that may help towards creating a revised paper version.
* * *
In the Abstract, please state the data used to test the two downscaling methods.

**We now state that we assess LOCA2 and STAR-ESDM *"…as compared to European Centre for medium-Range Weather Forecasts Reanalysis v5 (ERA5) and two observation-based data products (Livneh and nClimGrid-Daily)."***

In the Abstract, please also provide a typical spatial scale of the analysis that data and models (e.g., RCMs) can currently achieve.

**We now indicate that presently available downscaling products have ~5-km grid spacing. Further work is needed to determine the effective of resolution of the products given that they are derived from meteorological stations which are often more than 5 km apart.**

Additionally, the Abstract needs to be easy for a wide audience to understand. The sentence "…our results suggest that statistical downscaling techniques may be limited in their ability to resolve non-stationary hydrologic processes as compared to dynamical downscaling" is slightly ambiguous. Please make clear in the abstract that the word "dynamical" implies a more process-based approach. And "non-stationary"—is that referring to climate change? An Abstract should be broadly understandable in isolation from reading the full manuscript.

**We have revised the sentence to *"…our results indicate that statistical downscaling techniques may be limited in their ability to represent future physical processes that deviate from their historical behavior due to climate change."* We also indicate in the abstract that dynamical implies a more process-based approach and have removed the word "non-stationary" to make our abstract more broadly understandable.**

The Introduction is good, as it clearly differentiates between statistical and dynamical downscaling. Around line 54, it might help to give citations to example meteorological datasets that are used at the very fine scale, and of course as validation.

**We now cite Livneh et al., 2015 and Durre et al., 2022 as example datasets where suggested.**

The authors could expand a little more around line 61 regarding how there is no guarantee that statistical relationships between variables identified for current levels of atmospheric greenhouse gases will remain valid for future higher GHG concentrations. However a good start is testing performance for current GHG levels.

**We now expand on what was formerly Line 67: *"Additionally, statistical downscaling assumes that observed functional relationships will be preserved in the future (i.e., the stationarity assumption) despite climate change (Milly et al. 2008); however, there is no guarantee that historically derived statistical relationships will remain valid in the future."***

The paragraph starting "We compare" (line 117) is very helpful and differentiates well between dynamical downscaling (i.e. nested "RCMs", which contain physical process knowledge that is hopefully also valid for higher GHGs) and inferences from high-resolution meteorological data. The authors make good use of Tables (Table 1, CMIP6 models assessed and Table 2, "data" versus RCM). However, would it be possible to list in a Table, also measurement datasets. A combined Table might help.

**We now list the observation-based datasets (ERA5, Livneh dataset, and nClimGrid dataset) in Table 1 and provide their respective grid spacings.**

A further advantage of a carefully consider Table would be an opportunity to describe all the project names, ESM names, RCM names, datasets. I found myself continuously jumping between diagram captions and different parts of the text to fully understand what I was actually looking at. Some are not even defined – CONUS I guess is C for Canada, US for US? At first, reading the caption to Figure 1, I thought CONUS might be a dataset I had not heard of.

**We thank the Reviewer for this suggestion. We have combined all previous tables into a single table to make it easier for readers to reference datasets. Table 1 now lists raw CMIP6 GCMs, the observation-based datasets, the statistically downscaled datasets, and the NA-CORDEX dynamically downscaled datasets. We now clarify in the first instance of the phrase "contiguous US" that we will herein refer to it as CONUS.**

In isolation, Table 2 does not make full sense? If I've understood correctly (around line 125), then this Table is which RCMs (top row) are nested in which ESMs (left column). Correct? But I do not fully understand why ERA-Int is mentioned in the left column – surely the comparison would be between ERA-Int and each ESM/RCM combination? Maybe that is what the top row is getting at with "Analyzed" as each entry?

**In addition to the ESMs (left column), NA-CORDEX downscales the ERA-Interim dataset across all of the RCMs. A key benefit of this is that it allows for comparisons of RCM biases across a common dataset. For instance, if different ESM/RCM combinations were to be compared against each other, it would be difficult to assess the extent to which the differences are due to ESMs or RCMs. We now clarify this in the caption: "*In addition to global climate models, NA-CORDEX downscales ERA-Interim (top of left column) across different regional climate models (this allows for a comparison of downscaling across a common dataset).*"**

The aspect I like best about the analysis is the ability to compare the first column of Figure 1 (composite behaviour around a convective event) and Figure 2 (composite behaviour around frontal precipitation). To say the obvious, the curve shapes are very different. And critically, the relationship between the T and Precip curves is markedly different between the two. Please make sure that the reader is drawn to these differences in the main writing in the text (and/or discussion).

**We now draw the reader towards details describing features of temperature and precipitation curves (as well as their spatial expressions) during frontal precipitation events with a key sentence near the beginning of the paragraph starting around what was formerly Line 177: "*Our selection of frontal precipitation events show a very different relationship between temperature and precipitation as compared to convective precipitation.*"**

The next and important comparison, which again needs to come out very clearly from the paper is how actual datasets perform when compared to nested RCMs? Which Figure should I be looking at to compare against Figure1 and 2? Figure 4 is raw CMIP6 GCM data – so not the nested RCM outputs? ESM/RCM combinations are our best estimate of fine-scale meteorological behaviours, and so it should be them that are being tested the most against data?

The main message of the paper is that statistical downscaling does not capture co-variances between P and T as well as ESM/RCM i.e. process (or dynamical) downscaling? So again, the reader needs to see that really clearly too – which I guess implies comparing Figure 1 and 2 against Figure 9?

**The two comments above are related so we address them both here. As noted by the Reviewer, Figure 4 is the raw CMIP6 GCM results. Figure 10 is the ESM/RCM combination results that readers should compare to Figure 1 and 2 (as well as Figure 4). Our manuscript previously may not have been sufficiently clear to easily allow readers to compare across these figures. We have made concerted efforts to improve both the writing and quality of figures.**

The reader needs to be steered more clearly, leading to the four main messages:

- How do convective P-T relationships differ from frontal P-T relationships?
- Do statistical downscaling methods fail at this, as they cannot differentiate between the two necessarily.
- Do GCM/RCM combinations capture these differences better? And if so, then:
- What do GCM/RCM combinations project into the future?

**We thank the Reviewer for these comments. We have made concerted efforts to improve the quality of our writing and have made edits throughout the manuscript to more clearly steer readers towards our main conclusions. For instance, we now put in more references to specific figures in our Results section to more explicitly guide readers towards relevant results that support our key arguments.**

The presentation is poor in places. For instance, focusing on Figure 5, the left-column x-axis label needs to be marked as "days". Similarly, units are needed under the colourbars. The maps can be stretched by removing the white space of Canada and Mexico – this will make them easier to interpret. In many places, the diagrams can be enhanced in their appearance.

**We now mark units as suggested and have revised our figures to remove more of the white space of Canada and Mexico. We have also worked to improve the quality of the figures more broadly. With these changes, we believe that the figures are easier to interpret for readers.**

I am very happy to see the manuscript again, and in the meantime, I hope some of the suggestions and comments above are helpful.

**We thank the Reviewer for constructive comments that have helped improve our manuscript.**

---

## Author Comment (AC2)

RESPONSE TO REVIEWER #1 FOR *GEOSCIENTIFIC MODEL DEVELOPMENT*:
MANUSCRIPT EGUSPHERE-2024-1456
BY SEUNG H. BAEK, PAUL. A. ULLRICH, BO DONG, AND JIWOO LEE

We thank Reviewer #1 for thoughtful and constructive feedback. This Response to the Reviewer file provides a complete documentation of the changes that have been made in response to each individual comment. Reviewer's comments are shown in plain text. Authors' responses are shown in **bold**. Quotations from the revised manuscript are shown in ***bold italics***.

**Reviewer #1**

The paper deals with the evaluation of statistical and dynamical downscaling of the outputs of global climate models. The goals stated in the introduction are ambitious and interesting; however, the methods used have certain caveats, and the results do not bring any new findings, and the goals are not achieved. I recommend the rejection of the manuscript and encourage resubmission after the following comments are taken into account and the methodology is improved.

**We thank the Reviewer for the constructive comments. To our knowledge, it has not been observed elsewhere in the literature that statistical methods (at least the ones analyzed herein) diverge in estimated enhancement of frontal precipitation (an example of non-stationary process that is testable with just daily surface temperature and precipitation) that is otherwise robustly represented in dynamical methods (*e.g.,* raw GCMs and dynamical downscaling across 5 different regional climate models). We have made concerted efforts to highlight this novelty in our paper and more clearly state our new findings in our revised abstract.**

More detailed comments:

1. There are only a few references to related work (e.g., regarding uncertainties related to downscaling methods, evaluation of covariance structure in downscaled products, etc.), and the results obtained are not compared to previous studies.

   **We now reinforce the introduction section by citing several additional references. Giorgi (2018) and Lloyd et al. (2021) in particular comprehensively review the limitations of current RCMs (which we note in the manuscript). Regarding the comparison of our results to others, statistical downscaling methods to our knowledge have not yet been evaluated beyond single variable comparisons to observations (which we provide references for). The use of daily temperature and precipitation to examine key mechanisms in statistical downscaling products is a novel aspect of our paper not currently employed in the broader literature.**

2. The definitions of convective and frontal precipitation are rather simplistic. Only one event per year is selected, so only 21 days of each year are used for the analysis. This leads to only a limited amount of data analyzed. There is no discussion of possible other definitions or examples from the literature. Further, it is not quite clear how the events are selected. If the convective precipitation is defined using the annual maximum of air temperature, is it really the case that in every grid point the annual maximum of air temperature is followed by convective precipitation? Moreover, it is not clear how the "peak day" is chosen; further, "peak day" is only analyzed for observed datasets; it is not discussed whether it differs for the downscaling products and model outputs.

**We are very limited with the data we have over our disposal, as statistical downscaled products only provide daily temperature and precipitation outputs. While not comprehensive, the simplistic definitions are a strength of the paper insofar as they are required for uniform analyses that we can apply across statistical and dynamical downscaling. For convective precipitation in particular, we note that our definition is identical to that used in Zhang et al. (2023). We now stress this point in our manuscript:**

> ***"A central goal of our paper is to understand the representation of physical mechanisms in statistical downscaling products with only surface temperature and precipitation outputs (often the only two variables available with statistical downscaling). For this reason, we examine expected covariances between temperature and precipitation during convective and frontal precipitation events, including for the projection interval where the stationarity assumption may not hold."***

**We believe the co-evolution of temperature, precipitation, and moist static energy shown are strongly indicative of convective and frontal precipitation mechanisms, respectively. As the Reviewer mentions, it is true that only one event per year is selected (and thus only 21 days of each year are analyzed). However, these 21 days are the most likely of each year to capture convective precipitation (according to the intuition embedded in our definition), and we do this on a grid-by-grid basis to in reality analyze up to ~20 million+ (depending on resolution of climate products and though not necessarily independent) "convective precipitation events." While we do not expect every grid point of annual maximum of air temperature to be followed by convective precipitation, we do demonstrate this to be overwhelming the case, as each grid for the composite time series is weighted equally. In ERA5, the minimum precipitation anomaly for over 90% of the available grid points examined in our convective precipitation analysis occurs in day -2 to day 0. Therefore, it really is the case that for most (but admittedly not all) grid points, the annual maximum of air temperature is followed by convective precipitation. That is, by selecting for a very large sample size of events heavily biased for convective precipitation, we expect "noise" (*i.e.,* events not truly indicative of convective precipitation) to be negligible. Similar logic extends for frontal precipitation as well: in ERA5, the maximum precipitation anomaly for over 90% of the available grid points examined in our frontal precipitation analysis occurs in day +0 or day +1.**

We now clarify how "peak day" is chosen. For convective precipitation, we identify the day of highest daily maximum temperature (done grid-by-grid) for each year over 1980-2014. We then create a histogram of the number of times that the day of highest maximum temperature falls on a given day from 0 to 365 (thus days 0 – 365 are effectively histogram bins). Finally, we fit a discrete Fourier transform onto the histogram to identify the dominant frequency (*i.e.,* frequency corresponding to peak day) present in the data. We repeat similar steps but for day with the greatest drop in surface temperature for frontal precipitation. We now provide this clarification in the manuscript:

> *"To evaluate our method of identifying precipitation events, we (i) identify grid-by-grid the day of convective and frontal precipitation, respectively, for each year over 1980-2014; (ii) create histograms of the number of times that the day of convective or frontal precipitation falls between day 0 and day 365 of each calendar year (days 0 – 365 are thus effectively histogram bins); and (iii) fit a discrete Fourier transform onto the respective histogram to identify the dominant frequency (i.e., frequency corresponding to peak day) present in the data.*

As mentioned by the Reviewer, we previously only examined peak day for the observed dataset. We now also examine peak day for the 8 raw CMIP6 models (reproduced below as Figure R2R1). The results for the model clearly show that, as with ERA5, convective precipitation is dominant in the summer and frontal precipitation is dominant in winter (notwithstanding orographic rain in the western US). Given this agreement with observations and the fact that the Fourier transform is only conducted on surface temperature (and thus do not examine the joint evolution of surface temperature and precipitation), it is well expected that our analyses are appropriate across observations, raw GCMs, statistical downscaling products, and dynamical downscaling products. Statistical downscaling products, for instance, will only enhance agreement between GCMs and observations when it comes to just a single field. Local meteorology simulated in dynamical downscaling is not expected to interfere with the seasonality inherent in GCMs.

[Figure]

**Figure R2R1:** Same as Figure 3 of manuscript but for the 8 raw CMIP6 GCMs.

**3.** The data choice is not explained—why are only 8 CMIP6 GCMs used? For dynamical downscaling, the CMIP5-driven regional climate models are used, whereas for statistical downscaling, the CMIP6 GCMs are incorporated. In my opinion, the comparison of the results would be more informative if the same GCMs for both approaches were used. Moreover, there is no discussion of the choice of two specific statistical downscaling methods. It is claimed that they are "widely used" (l. 73). However, no references or examples are provided

**While we agree with the Reviewer that the comparison of the results would be more informative if the same GCMs for both approaches were used, the need for (i) same GCMs across LOCA2 and STAR-ESDM; (ii) availability of only CMIP5 models in dynamical downscaling efforts (*i.e.,* NA-CORDEX); and (iii) sufficient representation of a different regional climate models (we use five different RCMs) made this infeasible. The 8 CMIP6 models were chosen—admittedly somewhat arbitrarily—to balance the above-mentioned needs while also representing a sufficiently large ensemble size to show results that are robust across the CMIP6 ensemble (*i.e.,* any additional CMIP6 models would not appreciably change our results). We now note in the manuscript more clearly that the same 8 models are examined across the raw GCMs, LOCA2, and STAR-ESDM.**

**We nevertheless believe that 8 different lineage models (considered a large ensemble by most standards) is sufficient to minimize model-dependency. However, we verify this to be the case by performing an analysis similar to that of Figure 4 with three other CMIP6 models (CNRM-CM6-1, MPI-ESM-1-2-HAM, GISS-E2-2-G; provided below at Figure R2R2). Our results are therefore robust across 10+ different models. As noted by the Reviewer, the dynamical downscaling in NA-CORDEX uses CMIP5-driven regional climate models. However, we show that the behavior across five GCM-RCM combinations are highly consistent to those shown in the raw CMIP6 models.**

**Finally, we agree with the Reviewer that the specific choice of LOCA2 and STAR-ESDM was not well explained. We now provide 6 references to justify that LOCA2 and STAR-ESDM are widely-used. Equally importantly, we now state that the two techniques were *"selected to accompany the Fifth National Climate Assessment (NCA5; the preeminent guidance on national climate risks)"* to demonstrate that these two techniques are important operationally.**

[Figure]

**Figure R2R2:** 21-day composite time series (spatially averaged over CONUS domain) of **(a)** surface temperature anomalies (K) and **(b)** precipitation anomalies (mm/day) for (colored lines) CNRM-CM6-1, MPI-ESM-1-2-HAM, GISS-E2-2-G raw CMIP6 GCM and (solid black line) ERA5 data. Time series are centered around the day of convective precipitation and for the 1980-2014 period **(d-f)** Same as (a-c) but for frontal precipitation.

4. The covariance between air temperature and precipitation is discussed, but it is not calculated, or the values are not shown. The results are only shown in graphical form, which avoids quantitative evaluation. Moreover, the definitions of both convective precipitation and frontal precipitation, as used here, include the assumption of a temperature-precipitation relationship, making the results less informative. It would be very beneficial if the authors could come up with any quantitative evaluation of the covariances, enabling comparison of assessed methods in some overview figure/table.

**Covariance between temperature and precipitation during convective and frontal precipitation events are highly nonlinear. For instance, additional warm anomalies do not necessarily produce stronger convective precipitation; it is also the case that greater temperature gradients (*i.e.,* steeper cold fronts) do not necessarily produce stronger frontal precipitation. More broadly, surface temperature exerts rather weak influences and non-linear influences on precipitation on daily timescales (Pearson's correlation of daily 1979-2015 surface temperature and precipitation over 24 to 49°N and 125 to 67°W (approximating CONUS domain) is only 0.04.) We note that higher correlations are found at monthly or seasonal timescales (*e.g.,* Zhao and Khalil, 1993, Trenberth and Shea, 2005) but such timescales are not suitable for the purposes of our study.**

**We nevertheless address the Reviewer's greater concern regarding the lack of quantitative evaluation of the relationship between temperature and precipitation. We now provide kernel density estimates (KDE) of precipitation anomalies before convection (day -2) and after convection (day +2) for the 35-year composite of convective precipitation events. If there is no skill in our selection of convective precipitation (*i.e.,* events are randomly selected), precipitation anomalies before and after day +0 should be approximately equal. However, our analyses show that 97% of the CONUS grid points show higher precipitation anomalies at day +2 relative to day -2, showing a 97/3 split rather than a 50/50 split. Our KDE analyses show that the distribution of anomalies are significantly different (p<0.01) with a Kolmogorov-Smirnov test.**

**We perform similar analyses for our frontal precipitation analyses: 93% of the maximum precipitation during the 21-days analyzed in our 35-year composite of events occur on day +0 or day +1 (randomly selected events would see about 2/21 odds of this). Precipitation anomalies during day +0 and day +1 are significantly different (p<0.01) from the rest of the (randomly selected) 21-day window with a Kolmogorov-Smirnov test. We now include the below Figure R2R3 in our manuscript.**

[Figure]

**Figure R2R3: (left)** Kernel density estimates (KDE) of convective precipitation anomalies before convection (orange; day -2) and after convection (blue; day +2) for the 35-year composite of convective precipitation events. 97% of grid points during the 21-days analyzed show higher precipitation anomalies after convection. The two KDEs are significantly different (p<0.01) as determined by a Kolmogorov-Smirnov test. **(right)** Kernel density estimates of frontal precipitation anomalies on day +0 and day +1 (blue) and all other days of the 21-day window analyzed (orange; randomly sampled). 93% of the maximum precipitation occur on day +0 or day +1. The two KDEs are significantly different (p<0.01) as determined by a Kolmogorov-Smirnov test.

5. It is not explained why the authors concentrate specifically on frontal and convective precipitation. There are plenty of ways how to analyze the temperature-precipitation relationship, and the arguments for this specific choice should be provided.

**A central goal of our paper is to understand the representation of physical mechanisms in statistical downscaling products with only daily surface temperature and precipitation outputs. The limited availability of variables we have at our disposal (combined with the need to represent physical mechanisms) made frontal and convective precipitation suitable mechanisms for our evaluation. We nevertheless agree with the Reviewer's larger comment that argument for this specific choice was not provided in the manuscript and have addressed this shortcoming:**

> *"A central goal of our paper is to understand the representation of physical mechanisms in statistical downscaling products with only surface temperature and precipitation outputs (often the only two variables available with statistical downscaling). For this reason, we examine expected covariances between temperature and precipitation during convective and frontal precipitation events, including for the projection interval where the stationarity assumption may not hold."*

6. The conclusions summarized in the last section are very vague. For example, "statistical downscaling may not capture structural change to meteorological phenomena under non-stationarity" or "the dampening to be a spurious feature ... presumably from historical functional relationship and/or the non-stationarity assumption". One of the goals of the study formulated in the introduction was to study these issues in more detail, so, the conclusions of the study should be much stronger and more concrete.

**We thank the Reviewer for the feedback. We have made concerted efforts to remove vague conclusions and, in their stead, provide more concrete ones more consistent with the stated goals of the paper. We now provide a substantially revamped Conclusion section, including a new paragraph dedicated to stronger conclusions:**

> *"Our results are, to some extent, qualitatively intuitive: common statistical downscaling methods apply historical functional relationships to the future under the assumption that they will be preserved despite climate change. It is therefore somewhat expected that such techniques will provide lower skill for projections of non-stationary phenomena….Evaluation frameworks clearly demonstrating this to be the case has nevertheless proved elusive. Our work addresses this important gap by demonstrating that statistical downscaling methods diverge from estimated enhancement of frontal precipitation (an example of non-stationary process testable with just daily surface temperature and precipitation) where dynamical methods (e.g., raw GCMs and dynamical downscaling methods across 5 different regional climate models) do not…"*

7. ERA5 downscaled using dynamical downscaling - the references to NA-CORDEX (i.e., Mearns et al., 2017) nor the link to the NA-CORDEX data archive does not show any information about ERA5-driven simulations. From which source did the authors get the ERA5-driven simulations? The referred NA-CORDEX data include only ERA-Interim driven simulations.

**We meant ERA-Interim driven simulations (and not ERA5). We have corrected for this error.**

More specific/technical comments:

Figures, Figure captions: the term "composite" is not defined; precipitation anomalies shown in absolute values - this is not common, and the negative precipitation anomalies seem very strange; "MAE" and "SD" are not defined and explained; CONUS domain not defined; Fig. 4 - the parentheses are confusing, the caption needs to be reformulated to be more clear. Fig. 3 - for which dataset is it?

**We now define the term "composite" to refer to spatial averages over the CONUS domain. We also clarify in the figure captions that precipitation *anomalies* are shown relative to the 21-day window analyzed, resulting in both positive and negative values. We thank the Reviewer for the comment on not defining MAE. We now define MAE as mean absolute error in the figures. Upon re-reading the manuscript, we felt that the MAE and standard deviation (SD) provide somewhat repetitive information, so have opted to remove the standard deviation statistics from our paper. We have revised captions for Figure 3 and 4 to address the Reviewer's comments.**

Tables: the list of models should be accompanied by more information, e.g., horizontal resolution of the models, modeling centers, etc.

**We have edited the two tables to now include the names of modeling centers and the horizontal resolution of the models.**

l. 50-51: extremes are not physical processes

**We have removed "extremes" from the sentence.**

l. 58-62: the credibility of methods and relevancy of outputs are presented here to argue for the importance of physical consistency of climate change projections, even though the relevancy is not really important. The credibility based on physical consistency would be enough to introduce the covariance issue.

**We have removed relevancy as motivation in our sentence.**

Section 2: the observed datasets are referred to in a strange manner (e.g., "Livneh-unsplit" is not explained"); The explanation of the STAR-ESDM algorithm is not clear, mainly the term

"dynamic climatology"; The length of the studied periods - 35 years - seems rather strange, is not really common. Further, the fact that the reference period of 1980-2014 includes the years 2006-2014, which belong to the scenario simulation in the case of NA-CORDEX simulations. This should be at least mentioned, even though it presumably does not influence the results much.

**We have improved our description of the observed datasets, including for STAR-ESDM:** *"The STAR-ESDM algorithm first disaggregates observations and GCM outputs into four separate components: the long-term trend, climatological annual cycle, annually-varying annual cycle, and high frequency daily anomalies."* **Though somewhat arbitrary (and admittedly uncommon), we chose the 1980-2014 period to be roughly comparable to the 1979-2021 analyses conducted by Zhang et al. (2023) (note 2014 is the last day of LOCA2 historical). We nevertheless do not expect our results to be sensitive to the period chosen, given the large sample size of events analyzed. As suggested, we now mention that the years 2006-2014 belong to** *the scenario simulation for NA-CORDEX: "note that the years 2006-2014 fall under the RCP8.5 scenario for NA-CORDEX)."* **We also now mentioned this in the caption of Figure 10.**

l. 105: The spatial resolution of LOCA2 outputs is related to the spatial resolution of the underlying observed dataset, isn't it?

**Yes – we now note this in our manuscript:** *"The LOCA2 North American product uses an updated version of Livneh et al. 2015 with 6-km grid spacing as the training dataset (Pierce et al. 2021). Outputs from LOCA2 are also available at 6-km grid resolution."*

l. 119: "Ground truth" is a strange and inappropriate term. The uncertainties related to reference datasets should be discussed.

**We agree with the Review and now state:** *"Although observational climate datasets themselves have inherent uncertainties (such as from generation, sampling, or resolution; Zumwald et al. 2020), strong consistency across ERA5 and the two observation-based products reinforce the credibility of ERA5."* **We have moreover removed all other instances of the phrase "ground truth."**

l. 130: it is not clear how the information in the sentence "We therefore follow..." is implied from the previous sentence.

**We agree and have removed "therefore."**

Section 3: some of the terms used are confusing and uncommon, not well defined, e.g., "parallel time series", "post dynamical downscaling", "ensemble-mean differences" etc.

**We have made careful edits throughout Section 3 to clarify confusing and uncommon terms.**

---

## Author Response (AR2)

Reviewer #1

I would like to appreciate how the authors reacted and responded to my previous comments. All of them have been addressed appropriately. I would just like to ask the authors to consider the following: (line numbering relates to the file egusphere-2024-1456-ATC1.pdf):

**We thank the Reviewer and have made concerted efforts to address all of the below comments.**

l. 205-210: Please mention that the results are shown in Fig. 3

**We now mention that the moist static energy results are shown in Figure 2 (we presume that the Reviewer meant Figure 2 instead of Figure 3).**

Section 3.4: Please mention that the comparison between raw GCMs (CMIP6) and downscaled CMIP5 outputs might be influenced by the choice of the subsets being compared and that the results regarding the dynamical downscaling might be slightly different in case different model subsets are analyzed. For instance, some of the CMIP6 GCMs belong to those with higher climate sensitivity in comparison to CMIP5 (e.g., Meehl et al., 2020). Nevertheless, the conclusions of the study, mainly regarding the inability of statistical downscaling to preserve the T-Prec. relationship, are not dampened.

**We now state: "*Note that the comparison between the raw CMIP6 GCMs and downscaled CMIP5 outputs may be somewhat influenced by the specific subset of models, as some CMIP6 GCMs exhibit higher climate sensitivity in comparison to CMIP5 (e.g., Meehl et al., 2020).*"**

Fig. 1: Could you comment on the fact that precipitation remains high after the onset of convection (panel d)? In this regard, it might be relevant to comment on the spatial variability in precipitation after the onset of convection (panel f). Is the spatial variability similar for all ten days after the peak day?

**We now comment that precipitation remains high after the onset of convection:**
**"*Precipitation anomalies increase rapidly in the immediate days following [convection], coincident with rapid surface temperature anomaly decreases, then remains elevated after the onset of convection.*" We provide below in Figure R2R1 the spatial variability in precipitation after the onset of convection. The spatial variability does not appear to be similar for all nine days after the peak day.**

[Figure]

**Figure R2R1:** Spatial composite of precipitation anomalies (relative to the 21 day window analyzed) after convective precipitation.

Please check the units in the Figures. For example, in Fig. 11, the units for vertical axes are missing.

**We have doubled checked units in all the figures. Fig. 11, for instance, has units provided in the title.**

l. 300: "Over on" does not make sense; probably a typo.

**We have addressed this error (we presume that the Reviewer meant Line 200).**

l. 385 - 390: Please clarify what is the interpretation of MAE in connection to the P-T relationship.

**We now clarify in the captions that "[m]oist static energy increases until the precipitation event and rapidly decreases immediately afterwards as the atmosphere stabilizes."**

Meehl, G. A., Senior, C. A., Eyring, V., Flato, G., Lamarque, J. F., Stouffer, R. J., ... & Schlund, M. (2020). Context for interpreting equilibrium climate sensitivity and transient climate response from the CMIP6 Earth system models. Science Advances, 6(26), eaba1981.

**This is now cited.**

Reviewer #2

Thank you for asking me to re-review paper: "Evaluating downscaled products with expected hydroclimatic co-variances" by Baek et al.

I remain convinced that this paper makes an important contribution. It makes two important tests for downscaling of: (1) how well does it perform against key high-resolution datasets, and (2) what, in particular, happens when both T and P vary, so with an emphasis on post-storm behaviours in both variables, differentiated by frontal or convective behaviours. The latter focus is novel and important for those interested in likely meteorological effects after major storms.

I can see that the authors have worked hard to respond to review comments (both mine and the other reviewer), and this is really appreciated.

The final request at this stage is to improve slightly the presentation of the diagrams. Below are a few suggestions for the authors to consider.

**We thank the Reviewer and have made concerted efforts to improve the presentation of the diagrams.**

Figures 1,2, maybe stretch out the colorbars to be wider, as there is space. Possibly make the fonts of the tick labels slightly bigger? For the panels a, d, g, these are single variables, so maybe use black for the curves, and the vertical marker, as a dashed line.

**We have made the suggested changes to Figures 1 and 2.**

Figure 3 – this clearly requires some sort of alternative colourbar, of spread of values, to prevent the entire plot taking the same colour.

**We thank the Reviewer for the suggestion. After re-visiting the figure, the entire plot (for convective precipitation) appears to be taking the same color precisely because it is uniformly summer for the whole of the contiguous US (note for instance that there is much greater spread of values for the frontal precipitation plot). We have therefore opted to keep the original colorbar and spread of values.**

Figure 4 needs a legend inside the plot for the two colours. "P anomalies before convection", "P anomalies after convection" etc.

**We have added the legend.**

Figure 5 – According to Table 1, 8 ESMs are used. So there is likely room in one of the panels of Figure 5 to give a legend listing each of the ESMs. Or alternatively, you could place a legend stretched across the bottom of the multi-panel plot?

**We now place a legend stretched across the bottom of the multi-panel plot of Figure 5. We have also added a similar legend to Figures 8, 11, and 12.**

Figure 6 – I would help the reader in the caption by reminding that the LOCA2 data is a method to bias-correct different ESMs. So, then it is immediately clear that the different lines on (for instance) Figure 6a are the individual ESMs. Assuming this is the case, state "same colours as figure 5 for individual ESMs" (assuming a proper legend for the ESMs is given in Figure 5).

**We now remind readers in the caption that LOCA2 is a *"method that bias-corrects and downscales climate models." * We also now state that *"[c]olored lines indicate same models as in Figure 5."***

In the left-hand column of Figure 6, it might help putting a plot title of either "Convective Precipitation" or "Frontal Precipitation" across the top of each individual panel.

**We now indicate in the left-hand columns of Figures 6, 7, 9, and 10 that the top two panels are for convective precipitation and the bottom two panels are for frontal precipitation.**

The authors themselves might find other ways to tidy up the diagrams.

**We have made several adjustments, including those outlined above, to tidy up the diagrams.**

Other than issues of presentation, I think the paper is nearly ready for publication.

**We thank the Reviewer for comments that have improved the quality of our figures and the presentability of our paper.**